# ON THE INVERSION OF DEEP GENERATIVE MODELS

## ABSTRACT

Deep generative models (e.g. GANs and VAEs) have been developed quite extensively in recent years. Lately, there has been an increased interest in the inversion of such a model, i.e. given a (possibly corrupted) signal, we wish to recover the latent vector that generated it. Building upon sparse representation theory, we define conditions that rely only on the cardinalities of the hidden layer and are applicable to any inversion algorithm (gradient descent, deep encoder, etc.), under which such generative models are invertible with a unique solution. Importantly, the proposed analysis is applicable to any trained model, and does not depend on Gaussian i.i.d. weights. Furthermore, we introduce two layer-wise inversion pursuit algorithms for trained generative networks of arbitrary depth, where one of them is accompanied by recovery guarantees. Finally, we validate our theoretical results numerically and show that our method outperforms gradient descent when inverting such generators, both for clean and corrupted signals.

## 1 INTRODUCTION

In the past several years, deep generative models, e.g. Generative Adversarial Networks (GANs) (Goodfellow et al., 2014) and Variational Auto-Encoders (VAEs) (Kingma & Welling, 2013), have been greatly developed, leading to networks that can generate images, videos, and speech voices among others, that look and sound authentic to humans. Loosely speaking, these models learn a mapping from a random low-dimensional latent space to the training data distribution, obtained in an unsupervised manner.

Interestingly, deep generative models are not used only to generate arbitrary signals. Recent work rely on the inversion of these models to perform visual manipulations, compressed sensing, image interpolation, and more (Zhu et al., 2016; Bora et al., 2017; Simon & Aberdam, 2020). In this work, we study this inversion task. Formally, denoting the signal to invert by $\mathbf{y} \in \mathbb{R}^n$, the generative model as $G : \mathbb{R}^{n_0} \to \mathbb{R}^n$, and the latent vector as $\mathbf{z} \in \mathbb{R}^{n_0}$, we study the following problem:

$$\mathbf{z}^* = \arg\min_{\mathbf{z}} \frac{1}{2}\|G(\mathbf{z}) - \mathbf{y}\|_2^2, \tag{1}$$

where $G$ is assumed to be a feed-forward neural network.

The first question that comes to mind is *whether this model is invertible*, or equivalently, does Equation 1 have a unique solution? In this work, we establish theoretical conditions that guarantee the invertibility of the model $G$. Notably, the provided theorems are applicable to general non-random generative models, and do not depend on the chosen inversion algorithm.

Once the existence of a unique solution is recognized, the next challenge is to provide a recovery algorithm that is guaranteed to obtain the sought solution. A common and simple approach is to draw a random vector $\mathbf{z}$ and iteratively update it using gradient descent, opting to minimize Equation 1 (Zhu et al., 2016; Bora et al., 2017). Unfortunately, this approach has theoretical guarantees only in limited scenarios (Hand et al., 2018; Hand & Voroninski, 2019), since the inversion problem is generally non-convex. An alternative approach is to train an encoding neural network that maps images to their latent vectors (Zhu et al., 2016; Donahue et al., 2016; Bau et al., 2019; Simon & Aberdam, 2020); however, this method is not accompanied by any theoretical justification.

We adopt a third approach in which the generative model is inverted in an analytical fashion. Specifically, we perform the inversion layer-by-layer, similar to Lei et al. (2019). Our approach is based on the observation that every hidden layer is an outcome of a weight matrix multiplying a sparse

vector, followed by a ReLU activation. By utilizing sparse representation theory, the proposed algorithm ensures perfect recovery in the noiseless case and bounded estimation error in the noisy one. Moreover, we show numerically that our algorithm outperforms gradient descent in several tasks, including reconstruction of noiseless and corrupted images.

**Main contributions:** The contributions of this work are both theoretical and practical. We derive theoretical conditions for the invertiblity of deep generative models by ensuring a unique solution for the inversion problem defined in Equation 1. In short, these conditions rely on the growth of the non-zero elements of consecutive hidden layers by a factor of 2 for trained networks and by any constant greater than 1 for random models. Then, by leveraging the inherent sparsity of the hidden layers, we introduce a layerwise inversion algorithm with provable guarantees in the noiseless and noisy settings for fully-connected generators. To the best of our knowledge, this is the first work that provides such guarantees for general (non-random) models, addressing both the conceptual inversion and provable algorithms for solving Equation 1. Finally, we provide numerical experiments, demonstrating the superiority of our approach over gradient descent in various scenarios.

## 1.1 RELATED WORK

**Inverting deep generative models:** A tempting approach for solving Equation 1 is to use first order methods such as gradient descent. Even though this inversion is generally non-convex, the works in Hand & Voroninski (2019); Hand et al. (2018) show that if the weights are random then, under additional assumptions, no spurious stationary points exist, and thus gradient descent converges to the optimum. A different analysis, given in Latorre et al. (2019), studies the case of strongly smooth generative models that are near isometry. In this work, we study the inversion of general (non-random and non-smooth) ReLU activated generative networks, and provide a provable algorithm that empirically outperforms gradient descent. A close but different line of theoretical work analyzes the compressive sensing abilities of trained deep generative networks (Shah & Hegde, 2018; Bora et al., 2017); however, these works assume that an ideal inversion algorithm, solving Equation 1, exists. Different works Bojanowski et al. (2017); Wu et al. (2019) suggest training procedures that result with generative models that can be easily inverted. Nevertheless, in this work we do not assume anything on the training procedure itself, and only rely on the weights of the trained model.

**Layered-wise inversion:** The closest work to ours, and indeed its source of inspiration, is Lei et al. (2019), which proposes a novel scheme for inverting generative models. By assuming that the input signal was corrupted by bounded noise in terms of $\ell_1$ or $\ell_\infty$, they suggest inverting the model using linear programs layer-by-layer. That said, to assure a stable inversion, their analysis is restricted to cases where: (i) the network weights are Gaussian i.i.d. variables; (ii) the layers expand such that the number of non-zero elements in each layer is larger than the size of the entire layer preceding it; and (iii) that the last activation function is either ReLU or leaky-ReLU. Unfortunately, as mentioned in their work, these three assumptions often do not hold in practice. In this work, we do not rely on the distribution of the weights nor on the chosen activation function of the last layer. Furthermore, we relax the expansion assumption as to rely only on the expansion of the number of non-zero elements. This relaxation is especially needed in the last hidden layer, which is typically larger than the image size.

**Neural networks and sparse representation:** In the search for a profound theoretical understanding for deep learning, a series of papers suggested a connection between neural networks and sparse coding, by demonstrating that the forward pass of a neural network is in fact a pursuit for a multi-layer sparse representation (Papyan et al., 2017; Sulam et al., 2018; Chun & Fessler, 2019; Sulam et al., 2019; Romano et al., 2019; Xin et al., 2016). In this work, we expand this proposition by showing that the inversion of a generative model is based on sequential sparse coding steps.

## 2 THE GENERATIVE MODEL

**Notations:** We use bold uppercase letters to represent matrices, and bold lowercase letters to represent vectors. The vector $\mathbf{w}_j$ represents the $j$th column in the matrix $\mathbf{W}$. Similarly, the vector $\mathbf{w}_{i,j}$ represents the $j$th column in the matrix $\mathbf{W}_i$. The activation function ReLU is the entry-wise operator $\text{ReLU}(\mathbf{u}) = \max\{\mathbf{u}, \mathbf{0}\}$. We denote by $\text{spark}(\mathbf{W})$ the smallest number of columns in $\mathbf{W}$ that are linearly-dependent, and by $\|\mathbf{x}\|_0$ the number of non-zero elements in $\mathbf{x}$. The mutual

coherence of a matrix $\mathbf{W}$ is defined as: $\mu(\mathbf{W}) = \max_{i \neq j} \frac{|\mathbf{w}_i^T \mathbf{w}_j|}{\|\mathbf{w}_i\|_2 \|\mathbf{w}_j\|_2}$. Finally, we define $\mathbf{x}^{\mathcal{S}}$ and $\mathbf{W}_i^{\mathcal{S}}$ as the supported vector and the row-supported matrix according to the set $\mathcal{S}$, and denote by $\mathcal{S}^c$ the complementary set of $\mathcal{S}$.

**Problem Statement:** We consider a typical generative scheme $G : \mathbb{R}^{n_0} \to \mathbb{R}^n$ of the form:

$$
\begin{aligned}
\mathbf{x}_0 &= \mathbf{z}, \\
\mathbf{x}_{i+1} &= \text{ReLU}(\mathbf{W}_i \mathbf{x}_i), \text{ for all } i \in \{0, \dots, L-1\}, \\
G(\mathbf{z}) &= \phi(\mathbf{W}_L \mathbf{x}_L),
\end{aligned}
\tag{2}
$$

where $\mathbf{x}_i \in \mathbb{R}^{n_i}$, $\{\mathbf{x}_i\}_{i=1}^{L-1}$ are the hidden layers, $\mathbf{W}_i \in \mathbb{R}^{n_{i+1} \times n_i}$ are the weight matrices ($n_{L+1} = n$), $\mathbf{x}_0 = \mathbf{z} \in \mathbb{R}^{n_0}$ is the latent vector that is usually randomly selected from a normal distribution, $\mathbf{z} \sim \mathcal{N}(\mathbf{0}, \sigma^2 \mathbf{I}_{n_0})$, and $\phi$ is an invertible activation function, e.g. $\tanh$, sigmoid, or piece-wise linear.

Given a sample $\mathbf{x} = G(\mathbf{z})$, that was created by the generative model above, we aim to recover its latent vector $\mathbf{z}$. Note that each hidden vector in the model is produced by a ReLU activation, leading to hidden layers that are inherently sparse. This observation supports our approach to study this model utilizing sparse representation theory. In what follows, we use this observation to derive theoretical statements on the invertibility and the stability of this problem, and to develop pursuit algorithms that are guaranteed to restore the original latent vector.

## 3 INVERTIBILITY AND UNIQUENESS

We start by addressing this question: "Is this generative process invertible?". In other words, when given a signal that was generated by the model, $\mathbf{x} = G(\mathbf{z}^*)$, we know that a solution $\mathbf{z}^*$ to the inverse problem exists; however, can we ensure that this is the *only* one? Theorem 1 below (its proof is given in Appendix A) provides such guarantees, which are based on the sparsity level of the hidden layers and the spark of the weight matrices (see Section 2). Importantly, this theorem is not restricted to a specific pursuit algorithm; it can rather be used for any restoration method (gradient descent, deep encoder, etc.) to determine whether the recovered latent vector is the unique solution.

**Definition 1** (sub-spark). *Define the $s$-sub-spark of a matrix $\mathbf{W}$ as the minimal spark of any subset $\mathcal{S}$ of rows of cardinality $|\mathcal{S}| = s$, sub-spark$(\mathbf{W}, s) = \min_{|\mathcal{S}|=s} \text{spark}(\mathbf{W}^{\mathcal{S}})$.*

**Definition 2** (sub-rank). *Define the $s$-sub-rank of a matrix $\mathbf{W}$ as the minimal rank over any subset $\mathcal{S}$ of rows of cardinality $|\mathcal{S}| = s$, sub-rank$(\mathbf{W}, s) = \min_{|\mathcal{S}|=s} \text{rank}(\mathbf{W}^{\mathcal{S}})$.*

**Theorem 1** (Uniqueness). *Consider the generative scheme described in Equation 2 and a signal $\mathbf{x} = G(\mathbf{z}^*)$ with a corresponding set of representations $\{\mathbf{x}_i^*\}_{i=1}^L$ that satisfy:*

*(i)* $s_L = \|\mathbf{x}_L^*\|_0 < \frac{\text{spark}(\mathbf{W}_L)}{2}$.

*(ii)* $s_i = \|\mathbf{x}_i^*\|_0 < \frac{\text{sub-spark}(\mathbf{W}_i, s_{i+1})}{2}$, *for all* $i \in \{1, \dots, L-1\}$.

*(iii)* $n_0 = \text{sub-rank}(\mathbf{W}_0, s_1) \leq s_1$.

*Then, $\mathbf{z}^*$ is the unique solution to the inverse problem that meets these sparsity conditions.*

Theorem 1 is the first of its kind to provide uniqueness guarantees for general non-statistical weight matrices. Moreover, it only requires an expansion of the layer cardinalities as opposed to Huang et al. (2018); Hand & Voroninski (2019) and Lei et al. (2019) that require dimensionality expansion that often does not hold for the last layer (typically $n < n_L$).

A direct corollary of Theorem 1 is in the case of random matrices. In such case, the probability of heaving $n$ linearly dependent columns is essentially zero (Elad, 2010, Chapter 2). Hence, the conditions of Theorem 1 become:

$$
\text{(i)} \ \ s_L < \frac{n+1}{2}. \quad \text{(ii)} \ \ s_i < \frac{s_{i+1}+1}{2}. \quad \text{(iii)} \ \ s_1 \geq n_0.
\tag{3}
$$

In fact, since singular square matrices have Lebesgue measure zero, this corollary holds for almost all set of matrices.

In practice, to allow for a sufficient increase in the cardinalities of the hidden layers, their dimensions should expand as well, excluding the last layer. For example, if the dimensions of the hidden layers increase by a factor of 2, as long as the hidden layers preserve a constant percentage of non-zero elements, Theorem 1 holds almost surely. Notably, this is the common practice in various generative architectures, such as DC-GAN Radford et al. (2015) and PGAN Karras et al. (2017).

Nevertheless, in the random setting, we can further relax the above conditions by utilizing a theorem by Foucart & Rauhut (2013). This theorem considers a typical sparse representation model with a random dictionary and states that a sparse representation is unique as long as its cardinality is smaller than the signal dimension. Therefore, as presented in Theorem 2, in the random setting the cardinality across the layers need to grow only by a constant, i.e. $s_i < s_{i+1}$ and $s_L < n$.

**Theorem 2** (Uniqueness for Random Weight Matrices). *Assume that the weight matrices comprise of random independent and identically distributed entries (say Gaussian). If the representations of a signal $\mathbf{x} = G(\mathbf{z}^*)$ satisfy:*

*(i)* $s_L = \|\mathbf{x}_L\|_0 < n.$

*(ii)* $s_i = \|\mathbf{x}_i\|_0 < s_{i+1}$, *for all* $i \in \{1, \dots, L-1\}$.

*(iii)* $s_1 = \|\mathbf{x}_1\|_0 \geq n_0,$

*then, with probability 1, the inverse problem has a unique solution that meets these conditions.*

The above theorem states that to ensure a unique global minimum in the stochastic case, the number of nonzero elements should expand by only a single parameter. The proof of this theorem follows the same protocol as Theorem 1's proof, while replacing the spark-based uniqueness (Donoho & Elad, 2003) with Foucart & Rauhut (2013). As presented in Section 6.1, these conditions are very effective in predicting whether the generative process is invertible or not, regardless of the recovery algorithm used.

## 4 Pursuit Guarantees

In this section we provide an inversion algorithm supported by reconstruction guarantees for the noiseless and noisy settings. To reveal the potential of our approach, we first discuss the performance of an Oracle, in which the true supports of all the hidden layers are known, and only their values are missing. This estimation can be performed by a sequence of simple linear projections on the known supports. Note that already in the first step of estimating $\mathbf{x}_L$, we can realize the advantage of utilizing the inherent sparsity of the hidden layers. Here, the reconstruction error of the Oracle is proportional to $s_L = \|\mathbf{x}_L\|_0$, whereas solving a least square problem, as suggested in Lei et al. (2019), results with an error that is proportional to $n_L$. For more details see Appendix B.

---
**Algorithm 1** Layered Basis-Pursuit

**Input:** $\mathbf{y} = G(\mathbf{z}) + \mathbf{e} \in \mathbb{R}^n$, where $\|\mathbf{e}\|_2 \leq \epsilon$, and sparsity levels $\{s_i\}_{i=1}^L$.

**First step:** $\hat{\mathbf{x}}_L = \arg\min_{\mathbf{x}} \frac{1}{2} \left\| \phi^{-1}(\mathbf{y}) - \mathbf{W}_L \mathbf{x} \right\|_2^2 + \lambda_L \|\mathbf{x}\|_1$, with $\lambda_L = 2\ell\epsilon$.

Set $\hat{\mathcal{S}}_L = \text{Support}(\hat{\mathbf{x}}_L)$ and $\epsilon_L = \frac{(3+\sqrt{1.5})\sqrt{s_L}}{\min_j \|\mathbf{w}_{L,j}\|_2} \ell\epsilon$.

**General step:** For any layer $i = L-1, \dots, 1$ execute:

1. $\hat{\mathbf{x}}_i = \arg\min_{\mathbf{x}} \frac{1}{2} \left\| \hat{\mathbf{x}}_{i+1}^{\hat{\mathcal{S}}_{i+1}} - \mathbf{W}_i^{\hat{\mathcal{S}}_{i+1}} \mathbf{x} \right\|_2^2 + \lambda_i \|\mathbf{x}\|_1$, with $\lambda_i = 2\epsilon_{i+1}$.

2. Set $\hat{\mathcal{S}}_i = \text{Support}(\hat{\mathbf{x}}_i)$ and $\epsilon_i = \frac{(3+\sqrt{1.5})\sqrt{s_i}}{\min_j \left\| \mathbf{w}_{i,j}^{\hat{\mathcal{S}}_{i+1}} \right\|_2} \epsilon_{i+1}$.

**Final step:** Set $\hat{\mathbf{z}} = \arg\min_{\mathbf{z}} \frac{1}{2} \left\| \hat{\mathbf{x}}_1^{\hat{\mathcal{S}}_1} - \mathbf{W}_0^{\hat{\mathcal{S}}_1} \mathbf{z} \right\|_2^2$.

---

In what follows, we propose to invert the model by solving sparse coding problems layer-by-layer, while leveraging the sparsity of all the intermediate feature vectors. Specifically, Algorithm 1 describes a layered Basis-Pursuit approach, and Theorem 3 provides reconstruction guarantees for this algorithm. The proof of this theorem is given in Appendix C. In Corollary 1 we provide guarantees for this algorithm when inverting non-random generative models in the noiseless case.

**Definition 3** (Mutual Coherence of Submatrix). *Define $\mu_s(\mathbf{W})$ as the maximal mutual coherence of any submatrix of $\mathbf{W}$ with $s$ rows, $\mu_s(\mathbf{W}) = \max_{|\mathcal{S}|=s} \mu(\mathbf{W}^{\mathcal{S}})$.*

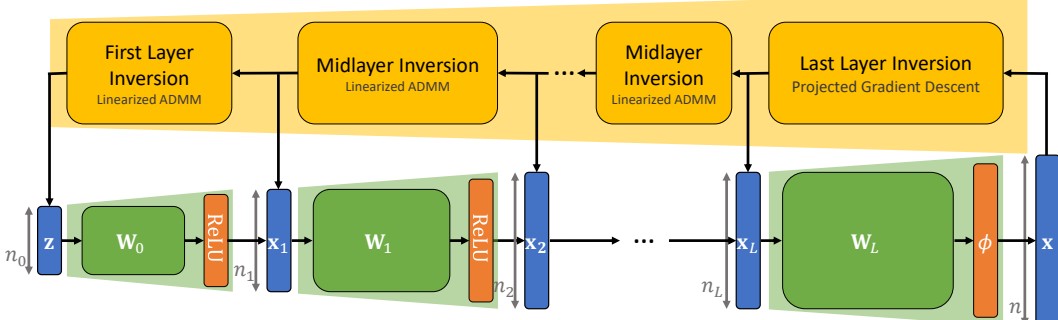

Figure 1: The Latent-Pursuit inverts the generative model layer-by-layer as described in Algorithm 3, and is composed of three steps: (1) last layer inversion by Algorithm 5; (2) midlayer inversions using Algorithm 2; and (3) first layer inversion via Algorithm 2 with the x-step replaced by Equation 10.

**Theorem 3** (Layered Basis-Pursuit Stability). *Suppose that $\mathbf{y} = \mathbf{x} + \mathbf{e}$, where $\mathbf{x} = G(\mathbf{z})$ is an unknown signal with known sparsity levels $\{s_i\}_{i=1}^{L}$, and $\|\mathbf{e}\|_2 \leq \epsilon$. Let $\ell$ be the Lipschitz constant of $\phi^{-1}$ and define $\epsilon_{L+1} = \ell\epsilon$. If in each midlayer $i \in \{1, \ldots, L\}$, $s_i < \frac{1}{3\mu_{s_{i+1}}(\mathbf{W}_i)}$, then,*

- *The support of $\hat{\mathbf{x}}_i$ is a subset of the true support, $\hat{\mathcal{S}}_i \subseteq \mathcal{S}_i$;*

- *The vector $\hat{\mathbf{x}}_i$ is the unique solution for the basis-pursuit;*

- *The midlayer's error satisfies $\|\hat{\mathbf{x}}_i - \mathbf{x}_i\|_2 < \epsilon_i$, where $\epsilon_i = \frac{(3+\sqrt{1.5})\sqrt{s_i}}{\min_j \left\|\mathbf{w}_{i,j}^{\hat{\mathcal{S}}_{i+1}}\right\|_2} \epsilon_{i+1}$.*

- *the recovery error on the latent space is upper bounded by*

$$\|\hat{\mathbf{z}} - \mathbf{z}\|_2 < \frac{\epsilon\ell}{\sqrt{\varphi}} \prod_{i=1}^{L} \frac{(3+\sqrt{1.5})\sqrt{s_j}}{\min_j \left\|\mathbf{w}_{i,j}^{\hat{\mathcal{S}}_{i+1}}\right\|_2}, \ \ where \ \varphi = \lambda_{\min}\left((\mathbf{W}_0^{\hat{\mathcal{S}}_1})^T \mathbf{W}_0^{\hat{\mathcal{S}}_1}\right) > 0. \quad (4)$$

**Corollary 1** (Layered Basis-Pursuit – Noiseless Case). *Let $\mathbf{x} = G(\mathbf{z})$ with sparsity levels $\{s_i\}_{i=1}^{L}$, and assume that $s_i < 1/3\mu_{s_{i+1}}(\mathbf{W}_i)$ for all $i \in \{1, \ldots, L\}$, and that $\varphi = \lambda_{\min}((\mathbf{W}_0^{\hat{\mathcal{S}}_1})^T \mathbf{W}_0^{\hat{\mathcal{S}}_1}) > 0$. Then Algorithm 1 recovers the latent vector $\hat{\mathbf{z}} = \mathbf{z}$ perfectly.*

## 5 THE LATENT-PURSUIT ALGORITHM

While Algorithm 1 provably inverts the generative model, it only uses the non-zero elements $\mathbf{x}_{i+1}^{\hat{\mathcal{S}}_{i+1}}$ to estimate the previous layer $\mathbf{x}_i$. Here we present the Latent-Pursuit algorithm, which expands the Layered Basis-Pursuit algorithm by imposing two additional constraints. First, the Latent-Pursuit sets inequality constraints, $\mathbf{W}_i^{\mathcal{S}_{i+1}^c}\mathbf{x}_i \leq \mathbf{0}$, that emerge from the ReLU activation. Second, recall that the ReLU activation constrains the midlayers to have nonnegative values, $\mathbf{x}_i \geq \mathbf{0}$. Furthermore, we refrain from inverting the activation function directly $\phi^{-1}$ since practically, this inversion might be unstable, e.g. when using $\tanh$. In what follows, we describe each of the three parts of the proposed algorithm: (i) the image layer; (ii) the middle layers; and (iii) the first layer.

Starting with the inversion of the last layer, i.e. the image layer, we need to solve

$$\mathbf{x}_L = \arg\min_{\mathbf{x}} \frac{1}{2}\|\mathbf{y} - \phi(\mathbf{W}_L\mathbf{x})\|_2^2 + \lambda_L \mathbf{1}^T\mathbf{x}, \ \ \mathrm{s.\,t.} \ \ \mathbf{x} \geq \mathbf{0}, \quad (5)$$

where $\mathbf{1}^T\mathbf{x}$ represents an $\ell_1$ regularization term under the nonnegative constraint. Assuming that $\phi$ is smooth and strictly monotonic increasing, this problem is a smooth convex function with separable constraints, and therefore, it can be solved using a projected gradient descent algorithm. In particular, we employ FISTA (Beck & Teboulle, 2009), as described in Algorithm 5 in Appendix D.

We move on to the middle layers, i.e. estimating $\mathbf{x}_i$ for $i \in \{1, \ldots, L-1\}$. Here, both the approximated vector and the given signal are assumed to result from a ReLU activation function. This leads us to the following problem:

$$\mathbf{x}_i = \arg\min_{\mathbf{x}} \frac{1}{2} \left\| \mathbf{x}_{i+1}^{\hat{\mathcal{S}}} - \mathbf{W}_i^{\hat{\mathcal{S}}} \mathbf{x} \right\|_2^2 + \lambda_i \mathbf{1}^T \mathbf{x}, \text{ s.t. } \mathbf{x} \geq \mathbf{0}, \mathbf{W}_i^{\hat{\mathcal{S}}^c} \mathbf{x} \leq \mathbf{0} \tag{6}$$

where $\hat{\mathcal{S}} = \hat{\mathcal{S}}_{i+1}$ is the support of the output of the layer to be inverted, and $\hat{\mathcal{S}}^c = \hat{\mathcal{S}}_{i+1}^c$ is its complementary. To solve this problem we introduce an auxiliary variable $\mathbf{a} = \mathbf{W}_i^{\mathcal{S}^c} \mathbf{x}$, leading to the following augmented Lagrangian form:

$$\min_{\mathbf{x}, \mathbf{a}, \mathbf{u}} \frac{1}{2} \left\| \mathbf{x}_{i+1}^{\mathcal{S}} - \mathbf{W}_i^{\mathcal{S}} \mathbf{x} \right\|_2^2 + \lambda_i \mathbf{1}^T \mathbf{x} + \frac{\rho_i}{2} \left\| \mathbf{a} - \mathbf{W}_i^{\mathcal{S}^c} \mathbf{x} + \mathbf{u} \right\|_2^2, \text{ s.t. } \mathbf{x} \geq \mathbf{0}, \mathbf{a} \leq \mathbf{0}. \tag{7}$$

This optimization problem could be solved using ADMM (Boyd et al., 2011), however, it would require inverting a matrix of size $n_i \times n_i$, which might be costly. Alternatively, we employ a more general method, called alternating direction *proximal* method of multipliers (Beck, 2017, Chapter 15), in which a quadratic proximity term, $\frac{1}{2} \left\| \mathbf{x} - \mathbf{x}^{(k)} \right\|_{\mathbf{Q}}$, is added to Equation 7. By setting

$$\mathbf{Q} = \alpha \mathbf{I} - \mathbf{W}_i^{\mathcal{S}^T} \mathbf{W}_i^{\mathcal{S}} + \beta \mathbf{I} - \rho_i \mathbf{W}_i^{\mathcal{S}^c T} \mathbf{W}_i^{\mathcal{S}^c}, \text{ with } \alpha + \beta \geq \lambda_{\max}(\mathbf{W}_i^{\mathcal{S}^T} \mathbf{W}_i^{\mathcal{S}} + \rho_i \mathbf{W}_i^{\mathcal{S}^c T} \mathbf{W}_i^{\mathcal{S}^c}), \tag{8}$$

we get that $\mathbf{Q}$ is positive semidefinite. This leads to the Linearized-ADMM algorithm in Algorithm 2 which is guaranteed to converge to the optimal solution of Equation 7 (see details in Appendix D).

We now recovered all the hidden layers, and only the latent vector $\mathbf{z}$ is left to be estimated. For this inversion step we adopt a MAP estimator utilizing the fact that $\mathbf{z}$ is drawn from a normal distribution:

$$\mathbf{z} = \arg\min_{\mathbf{z}} \frac{1}{2} \left\| \mathbf{x}_1^{\mathcal{S}} - \mathbf{W}_0^{\mathcal{S}} \mathbf{z} \right\|_2^2 + \frac{\gamma}{2} \left\| \mathbf{z} \right\|_2^2, \text{ s.t. } \mathbf{W}_0^{\mathcal{S}^c} \mathbf{z} \leq \mathbf{0}, \tag{9}$$

with $\gamma > 0$. This problem can be solved by the Linearized-ADMM algorithm described above (see details in Appendix D), except for the update of $\mathbf{x}$ in Algorithm 2, which becomes:

$$\mathbf{z}^{(k+1)} \leftarrow \frac{1}{\alpha + \beta + \gamma} \left( (\alpha + \beta) \mathbf{z}^{(k)} - \mathbf{W}_0^{\mathcal{S}^T} (\mathbf{W}_0^{\mathcal{S}} \mathbf{z}^{(k)} - \mathbf{x}_1^{\mathcal{S}}) - \rho_1 \mathbf{W}_0^{\mathcal{S}^c T} (\mathbf{W}_0^{\mathcal{S}^c} \mathbf{z}^{(k)} - \mathbf{a}^{(k)} - \mathbf{u}^{(k)}) \right). \tag{10}$$

Once the latent vector $\mathbf{z}$ and all the hidden layers $\{\mathbf{x}_i\}_{i=1}^L$ are recovered, we propose an optional step to improve the final estimation. In this step, which we refer to as debiasing, we freeze the recovered supports and only optimize over the non-zero values in an end-to-end fashion. This is equivalent to computing the Oracle, only here the supports are not known, but rather estimated using the proposed pursuit. Algorithm 3 provides a short description of the entire proposed inversion method.

---

**Algorithm 2** Latent Pursuit: Midlayer Inversion

**Initialization:** $\mathbf{x}^{(0)} \in \mathbb{R}^{n_i}$, $\mathbf{u}^{(0)}, \mathbf{a}^{(0)} \in \mathbb{R}^{s_{i+1}}$, $\rho_i > 0$, and $\alpha, \beta$ satisfying Equation 8.
**Until converged:** for $k = 0, 1, \ldots$ execute:

1. $\mathbf{x}^{(k+1)} \leftarrow \text{ReLU} \left( \mathbf{x}^{(k)} - \frac{1}{\alpha+\beta} \mathbf{W}_i^{\mathcal{S}^T} (\mathbf{W}_i^{\mathcal{S}} \mathbf{x}^{(k)} - \mathbf{x}_{i+1}^{\mathcal{S}}) - \frac{\rho_i}{\alpha+\beta} \mathbf{W}_i^{\mathcal{S}^c T} (\mathbf{W}_i^{\mathcal{S}^c} \mathbf{x}^{(k)} - \mathbf{a}^{(k)} - \mathbf{u}^{(k)}) - \frac{\lambda_i}{\alpha+\beta} \right)$.

2. $\mathbf{a}^{(k+1)} \leftarrow -\text{ReLU} \left( \mathbf{u}^{(k)} - \mathbf{W}_i^{\mathcal{S}^c} \mathbf{x}^{(k+1)} \right)$.

3. $\mathbf{u}^{(k+1)} \leftarrow \mathbf{u}^{(k)} + \mathbf{a}^{(k+1)} - \mathbf{W}_i^{\mathcal{S}^c} \mathbf{x}^{(k+1)}$.

**Algorithm 3** The Latent-Pursuit Algorithm

**Initialization:** Set $\lambda_i > 0$ and $\rho_i > 0$.
**First step:** Estimate $\mathbf{x}_L$, i.e. solve Equation 5 using Algorithm 5.
**Middle step:** For layers $i = L - 1, \ldots, 1$, estimate $\mathbf{x}_i$ using Algorithm 2.
**Final step:** Estimate $\mathbf{z}$ using Algorithm 2 but with the $x$-step of Equation 10.
**Debiasing (optional):** Set $\mathbf{z} \leftarrow \arg\min_{\mathbf{z}} \frac{1}{2} \left\| \mathbf{y} - \phi \left( \left( \prod_{i=L}^0 \mathbf{W}_i^{\hat{\mathcal{S}}_{i+1}} \right) \mathbf{z} \right) \right\|_2^2$.

---

## 6 Numerical Experiments

We demonstrate the effectiveness of our approach through numerical experiments, where our goal is twofold. First, we study random generative models and show the ability of the uniqueness claim above (Theorem 2) to predict when both gradient descent and our approach fail to invert $G$ as the inversion is not unique. In addition, we show that in these random networks and under the conditions of Corollary 1, the latent vector is perfectly recovered by both the Layered Basis-Pursuit and the Latent-Pursuit algorithm. Our second goal is to demonstrate the advantage of Latent-Pursuit over gradient descent for trained generative models, in two settings: noiseless and image inpainting.

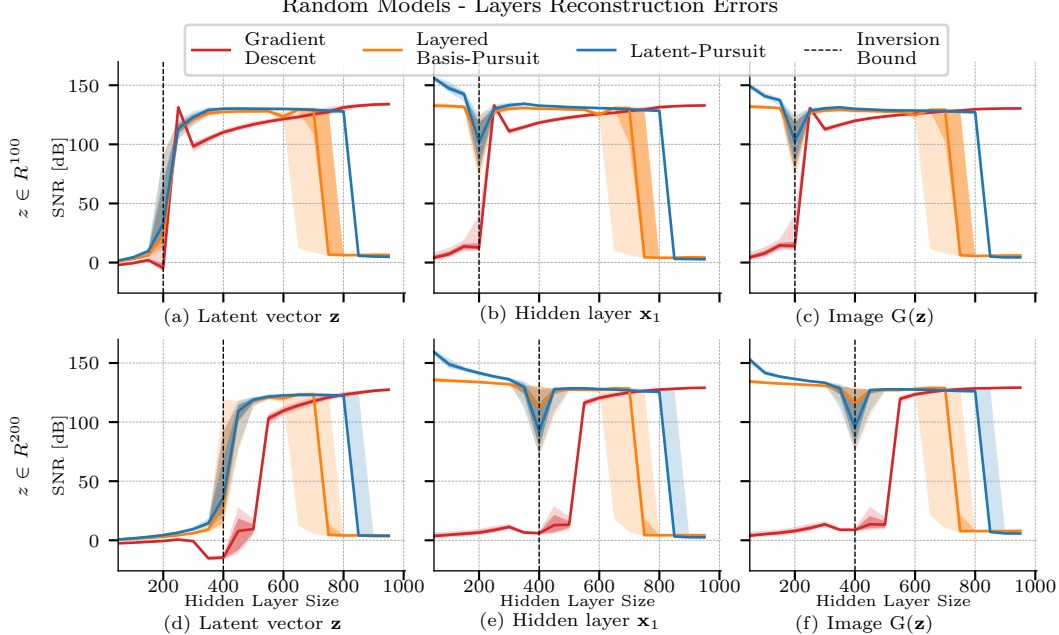

Figure 2: Gaussian iid weights: Recovery errors as a function of the hidden-layer size ($n_1$), where the image space is 625. Subfigures (a)-(c) correspond to $\mathbf{z} \in \mathbb{R}^{100}$ and (d)-(f) to $\mathbf{z} \in \mathbb{R}^{200}$. These results support Theorem 2 stating that to guarantee a unique solution, the hidden layer cardinality $s_1 \approx \frac{n_1}{2}$ should be larger than the latent vector space and smaller than the image space. Moreover, it supports Corollary 1 by showing that under the non-zero expansion condition, both Layered Basis-Pursuit and Latent-Pursuit (Algorithms 1 and 3) recover the original latent vector perfectly.

## 6.1 RANDOM WEIGHTS

First, we validate the above theorems on random generative models, by considering a framework similar to Huang et al. (2018) and Lei et al. (2019). Here, the generator is composed of two layers:

$$\mathbf{x} = G(\mathbf{z}) = \tanh(\mathbf{W}_2 \operatorname{ReLU}(\mathbf{W}_1 \mathbf{z})), \tag{11}$$

where the dimensions of the network are $n = 625$, $n_1$ varies between 50 to 1000 and $n_0 \in \{100, 200\}$. The weight matrices $\mathbf{W}_1$ and $\mathbf{W}_2$ are drawn from an iid Gaussian distribution. We generate 512 signals by feeding the generator with latent vectors from a Gaussian distribution, and then test the performance of the inversion of these signals in terms of SNR for all the layers, using gradient descent, Layered Basis-Pursuit (Algorithm 1), and Latent-Pursuit (Algorithm 3). For gradient descent, we use the smallest step-size from $\{1e-1, 1e0, 1e1, 1e2, 1e3, 1e4\}$ for $10,000$ steps that resulted with a gradient norm smaller than $1e-9$. For Layered Basis-Pursuit we use the best $\lambda_1$ from $\{1e-5, 7e-6, 3e-6, 1e-6, 0\}$, and for Latent-Pursuit, we use $\lambda_1 = 0$, $\rho = 1e-2$ and $\gamma = 0$. In Layered Basis-Pursuit and Latent-Pursuit we preform a debiasing step in a similar manner to gradient descent. Figure 2 marks median results in the central line, while the ribbons show $90\%$, $75\%$, $25\%$, and $10\%$ quantiles. In these experiments the sparsity level of the hidden layer is approximately $50\%$, $s_1 = \|\mathbf{x}_1\|_0 \approx \frac{n_1}{2}$, due to the weights being random. In what follows, we split the analysis of Figure 2 to three segments. Roughly, these segments are $s_1 < n_0$, $n_0 < s_1 < n$, and $n < s_1$ as suggested by the theoretical results given in Theorem 2 and Corollary 1.

In the first segment, Figure 2 shows that all three methods fail. Indeed, as suggested by the uniqueness conditions introduced in Theorem 2, when $s_1 < n_0$, the inversion problem of the first layer does not have a unique global minimizer. The dashed vertical line in Figure 2 marks the spot where $\frac{n_1}{2} = n_0$. Interestingly, we note that the conclusions in (Huang et al., 2018; Lei et al., 2019), suggesting that large latent spaces cause gradient descent to fail, are imprecise and valid only for fixed hidden layer size. This can be seen by comparing $n_0 = 100$ to $n_0 = 200$. As a direct outcome of our uniqueness study and as demonstrated in Figure 2, gradient descent (and any other algorithm) fails when the ratio between the cardinalities of the layers is smaller than 2. Nevertheless, Figure 2

exposes an advantage for using our approach over gradient descent. Note that our methods successfully invert the model for all the layers that follow the layer for which the sparsity assumptions do not hold, and fail only past that layer, since only then uniqueness is no longer guaranteed. However, since gradient descent starts at a random location, all the layers are poorly reconstructed.

For the second segment, we recall Theorem 3 and in particular Corollary 1. There we have shown that Layered Basis-Pursuit and Latent-Pursuit are guaranteed to perfectly recover the latent vector as long as the cardinality of the midlayer $s_1 = \|\mathbf{x}_1\|_0$ satisfies $n_0 \leq s_1 \leq 1/3\mu(\mathbf{W}_1)$. Indeed, Figure 2 demonstrates the success of these two methods even when $s_1 \approx \frac{n_1}{2}$ is greater than the worst-case bound $1/3\mu(\mathbf{W}_1)$. Moreover, this figure validates that Latent-Pursuit, which leverages additional properties of the signal, outperforms Layered Basis-Pursuit, especially when $s_1$ is large. Importantly, while the analysis in Lei et al. (2019) suggests that $n$ has to be larger than $n_1$, in practice, all three methods succeed to invert the signal even when $n_1 > n$. This result highlights the strength of the proposed analysis that leans on the cardinality of the layers rather than their size.

We move on to the third and final segment, where the size of hidden layer is significantly larger than the dimension of the image. Unfortunately, in this scenario the layer-wise methods fail, while gradient descent succeeds. Note that, in this setting, inverting the last layer solely is an ambitious (actually, impossible) task; however, since gradient descent solves an optimization problem of a much lower dimension, it succeeds in this case as well. This experiment and the accompanied analysis suggest that a hybrid approach, utilizing both gradient descent and the layered approach, might be of interest. We defer a study of such an approach for future work.

## 6.2 TRAINED NETWORK

To demonstrate the practical contribution of our work, we experiment with a generative network trained on the MNIST dataset. Our architecture is composed of fully connected layers of sizes 20, 128, 392, and finally an image of size $28 \times 28 = 784$. The first two layers include batch-normalization[1] and a ReLU activation function, whereas the last one includes a piecewise linear unit (Nicolae, 2018). We train this network in an adversarial fashion using a fully connected discriminator and spectral normalization (Miyato et al., 2018). We should note that images produced by fully connected models are typically not as visually appealing as ones generated by convolutional architectures. However, since the theory provided here focuses on fully connected models, this setting was chosen for the experimental section, similar to other previous work (Huang et al., 2018; Lei et al., 2019) that study the inversion process.

**Network inversion:** We start with the noiseless setting and compare the Latent-Pursuit algorithm to the Oracle (which knows the exact support of each layer) and to gradient descent. To invert a signal and compute its reconstruction quality, we first invert the entire model and estimate the latent vector. Then, we feed this vector back to the model to estimate the hidden representations and the reconstructed image. For our algorithm we use $\rho = 1e-2$ for all layers and $10,000$ iterations of debiasing. For gradient-descent run, we use $10,000$ iterations, momentum of $0.9$ and a step size of $1e-1$ that gradually decays to assure convergence. Overall, we repeat this experiment $512$ times.

Figure 3a demonstrates the reconstruction error of the latent vector. First, we observe that the performance of our inversion algorithm is on par with those of the Oracle. Moreover, not only does our approach performs much better than gradient descent, but in many experiments the latter fails utterly. In Appendix E.1 we provide reconstruction error for all the layers followed by image samples.

A remark regarding the run-time of these algorithms is in place. Using an Nvidia 1080Ti GPU, the proposed Latent-Pursuit algorithm took approximately $15$ seconds per layer to converge for a total of approximately $75$ seconds to complete, including the debiasing step, for all $512$ experiments. On the other hand, gradient-descent took approximately $30$ seconds to conclude.

**Image inpainting:** We continue our experiments with image inpainting, i.e. inverting the network and reconstructing a clean signal when only some of its pixels are known. First, we apply a random mask in which $45\%$ of the pixels are randomly concealed. Since the number of known pixels is still larger than the number of non-zero elements in the layer preceding it, our inversion algorithm usually reconstructs the image successfully as suggested by Figure 3b. In this experiment, we perform slightly worse than the Oracle, which is not surprising considering the information disparity between

---

[1]Note that after training, batch-normalization is a simple linear operation.

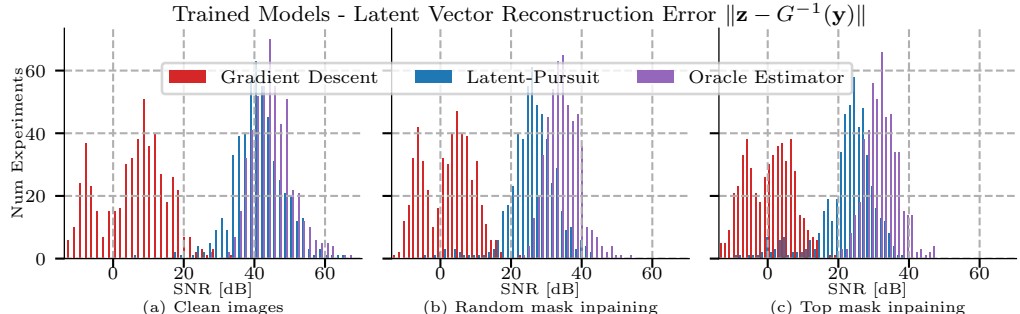

Figure 3: Trained 3 layers model: Reconstruction error of the latent vector $z$ in $512$ experiments. Latent-Pursuit clearly outperforms gradient descent.

the two. As for gradient descent, we see similar results to the ones received in the non-corrupted setting. Appendix E.2 provides image samples and reconstruction comparison across all layers. Finally, we repeat the above experiment using a deterministic mask that conceals the upper $\sim 45\%$ of each image (13 out of 28 rows). The results of this experiment, which are provided in Figure 3c and Appendix E.3, lead to similar conclusions as in the previous experiment. Indeed, since the model contains fully connected layers, we expect the last two experiments to show comparable results.

## 7    CONCLUSIONS

In this paper we have introduced a novel perspective regarding the inversion of deep generative networks and its connection to sparse representation theory. Building on this, we have proposed novel invertibility guarantees for such a model for both random and trained networks. We have accompanied our analysis by novel pursuit algorithms for this inversion and presented numerical experiments that validate our theoretical claims and the superiority of our approach compared to the more classic gradient descent. We believe that the insights underlining this work could lead to a broader activity which further improves the inversion of these models in a variety of tasks.

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

## A    THEOREM 1: PROOF

*Proof.* The main idea of the proof is to show that under the conditions of Theorem 1 the inversion task at every layer $i \in \{1, \ldots, L+1\}$ has a unique global minimum. For this goal we utilize the well-known uniqueness guarantee from sparse representation theory.

**Lemma 1** (Sparse Representation - Uniqueness Guarantee Donoho & Elad (2003); Elad (2010)). *If a system of linear equations $\mathbf{y} = \mathbf{W}\mathbf{x}$ has a solution $\mathbf{x}$ satisfying $\|\mathbf{x}\|_0 < \text{spark}(\mathbf{W})/2$, then this solution is necessarily the sparset possible.*

Using the above Lemma, we can conclude that if $\mathbf{x}_L$ obeys $\|\mathbf{x}_L\|_0 = s_L < \text{spark}(\mathbf{W}_L)/2$, then $\mathbf{x}_L$ is the *unique* vector that has at most $s_L$ nonzeros, while satisfying the equation $\phi^{-1}(\mathbf{x}) = \mathbf{W}_L \mathbf{x}_L$.

Moving on to the previous layer, we can employ again the above Lemma for the supported vector $\mathbf{x}_L^{\mathcal{S}_L}$. This way, we can ensure that $\mathbf{x}_{L-1}$ is the unique $s_{L-1}$-sparse solution of $\mathbf{x}_L^{\mathcal{S}_L} = \mathbf{W}_{L-1}^{\mathcal{S}_L} \mathbf{x}_{L-1}$ as long as

$$s_{L-1} = \|\mathbf{x}_{L-1}\|_0 < \frac{\text{spark}(\mathbf{W}_{L-1}^{\mathcal{S}_L})}{2}. \tag{12}$$

However, the condition $s_{L-1} = \|\mathbf{x}_{L-1}\|_0 < \frac{\text{sub-spark}(\mathbf{W}_{L-1}, s_L)}{2}$ implies that the above necessarily holds. This way we can ensure that each layer $i, i \in \{1, \ldots, L-1\}$ is the unique sparse solution.

Finally, in order to invert the first layer we need to solve $\mathbf{x}_1^{\mathcal{S}_1} = \mathbf{W}_0^{\mathcal{S}_1} \mathbf{z}$. If $\mathbf{W}_0^{\mathcal{S}_1}$ has full column-rank, this system either has no solution or a unique one. In our case, we do know that a solution exists, and thus, necessarily, it is unique. A necessary but insufficient condition for this to be true is $s_1 \geq n_0$. The additional requirement $\text{sub-rank}(\mathbf{W}_0, s_1) = n_0 \leq s_1$ is sufficient for $\mathbf{z}$ to be the unique solution, and this concludes the proof.

□

## B    THE ORACLE ESTIMATOR

The motivation for studying the recovery ability of the Oracle is that it can reveal the power of utilizing the inherent sparsity of the feature maps. Therefore, we analyze the layer-wise Oracle estimator described in Algorithm 4, which is similar to the layer-by-layer fashion we adopt in both the Layered Basis-Pursuit (Algorithm 1) and in the Latent-Pursuit (Algorithm 3). In this analysis we assume that the contaminating noise is white additive Gaussian.

The noisy signal $\mathbf{y}$ carries an additive noise with energy proportional to its dimension, $\sigma^2 n$. Theorem 4 below suggests that the Oracle can attenuate this noise by a factor of $\frac{n_0}{n}$, which is typically much smaller than 1. Moreover, the error in each layer is proportional to its cardinality $\sigma^2 s_i$. These results are expected, as the Oracle simply projects the noisy signal on low-dimensional subspaces of known

---

**Algorithm 4** The Layered-Wise Oracle

---

**Input:** $\mathbf{y} = G(\mathbf{z}) + \mathbf{e} \in \mathbb{R}^n$, and *supports of each layer* $\{\mathcal{S}_i\}_{i=1}^{L}$.

**First step:** $\hat{\mathbf{x}}_L = \arg\min_{\mathbf{x}} \frac{1}{2} \left\| \phi^{-1}(\mathbf{y}) - \bar{\mathbf{W}}_L \mathbf{x} \right\|_2^2$, where $\bar{\mathbf{W}}_L$ is the column supported matrix $\mathbf{W}_L[:, \mathcal{S}_L]$.

**Intermediate steps:** For any layer $i = L-1, \ldots, 1$, set $\hat{\mathbf{x}}_i = \arg\min_{\mathbf{x}} \frac{1}{2} \left\| \hat{\mathbf{x}}_{i+1}^{\mathcal{S}_{i+1}} - \bar{\mathbf{W}}_i \mathbf{x} \right\|_2^2$, where $\bar{\mathbf{W}}_i$ is the row and column supported matrix $\mathbf{W}_i[\mathcal{S}_{i+1}, \mathcal{S}_i]$.

**Final step:** Set $\hat{\mathbf{z}} = \arg\min_{\mathbf{z}} \frac{1}{2} \left\| \hat{\mathbf{x}}_1^{\mathcal{S}_1} - \mathbf{W}_0^{\mathcal{S}_1} \mathbf{z} \right\|_2^2$.

---

dimension. That said, this result reveals another advantage of employing the sparse coding approach over solving least squares problems, as the error can be proportional to $s_i$ rather than to $n_i$.

**Theorem 4** (The Oracle). *Given a noisy signal* $\mathbf{y} = G(\mathbf{z}) + \mathbf{e}$, *where* $\mathbf{e} \sim \mathcal{N}(\mathbf{0}, \sigma^2 \mathbf{I})$, *and assuming known supports* $\{\mathcal{S}_i\}_{i=1}^{L}$, *the recovery errors satisfy* [2]:

$$\frac{\sigma^2}{\prod_{j=i}^{L} \lambda_{\max}(\bar{\mathbf{W}}_j^T \bar{\mathbf{W}}_j)} s_i \leq \mathbb{E} \left\| \hat{\mathbf{x}}_i - \mathbf{x}_i \right\|_2^2 \leq \frac{\sigma^2}{\prod_{j=i}^{L} \lambda_{\min}(\bar{\mathbf{W}}_j^T \bar{\mathbf{W}}_j)} s_i, \tag{13}$$

*for* $i \in \{1, \ldots, L\}$, *where* $\bar{\mathbf{W}}_i$ *is the row and column supported matrix,* $\mathbf{W}_i[\mathcal{S}_{i+1}, \mathcal{S}_i]$. *The recovery error bounds for the latent vector are similarly given by:*

$$\frac{\sigma^2}{\prod_{j=0}^{L} \lambda_{\max}(\bar{\mathbf{W}}_j^T \bar{\mathbf{W}}_j)} n_0 \leq \mathbb{E} \left\| \hat{\mathbf{z}} - \mathbf{z} \right\|_2^2 \leq \frac{\sigma^2}{\prod_{j=0}^{L} \lambda_{\min}(\bar{\mathbf{W}}_j^T \bar{\mathbf{W}}_j)} n_0. \tag{14}$$

*Proof.* Assume $\mathbf{y} = \mathbf{x} + \mathbf{e}$ with $\mathbf{x} = G(\mathbf{z})$, then the Oracle for the $L$th layer is $\hat{\mathbf{x}}_L^{\mathcal{S}} = \bar{\mathbf{W}}_L^{\dagger} \mathbf{y}$. Since $\mathbf{y} = \bar{\mathbf{W}}_L \mathbf{x}_L^{\mathcal{S}} + \mathbf{e}$, we get that $\hat{\mathbf{x}}_L^{\mathcal{S}} = \mathbf{x}_L^{\mathcal{S}} + \tilde{\mathbf{e}}_L$, where $\tilde{\mathbf{e}}_L = \bar{\mathbf{W}}_L^{\dagger} \mathbf{e}$, and $\tilde{\mathbf{e}}_L \sim \mathcal{N}(\mathbf{0}, \sigma^2 (\bar{\mathbf{W}}_L^T \bar{\mathbf{W}}_L)^{-1})$. Therefore, using the same proof technique as in Aberdam et al. (2019), the upper bound on the recovery error in the $L$th layer is:

$$\mathbb{E} \left\| \hat{\mathbf{x}}_L - \mathbf{x}_L \right\|_2^2 = \sigma^2 \operatorname{trace}((\bar{\mathbf{W}}_L^T \bar{\mathbf{W}}_L)^{-1}) \leq \sigma^2 \frac{s_L}{\lambda_{\min}(\bar{\mathbf{W}}_L^T \bar{\mathbf{W}}_L)}. \tag{15}$$

Using the same approach we can derive the lower bound by using the largest eigenvalue of $\bar{\mathbf{W}}_L^T \bar{\mathbf{W}}_L$. In a similar fashion, we can write $\hat{\mathbf{x}}_i^{\mathcal{S}} = \mathbf{x}_i^{\mathcal{S}} + \tilde{\mathbf{e}}_i$ for all $i \in \{0, \ldots, L-1\}$, where $\tilde{\mathbf{e}}_i = \mathbf{A}_{[i,L]} \mathbf{e}$ and $\mathbf{A}_{[i,L]} \triangleq \bar{\mathbf{W}}_i^{\dagger} \bar{\mathbf{W}}_{i+1}^{\dagger} \cdots \bar{\mathbf{W}}_L^{\dagger}$. Therefore, the upper bound for the recovery error in the $i$th layer

---

[2]For simplicity we assume here that $\phi$ is the identity function.

becomes:

$$
\begin{aligned}
\mathbb{E} \left\| \hat{\mathbf{x}}_i - \mathbf{x}_i \right\|_2^2 &= \mathbb{E} \left\| \mathbf{A}_{[i,L]} \mathbf{e} \right\|_2^2 \\
&= \sigma^2 \operatorname{trace} \left( \mathbf{A}_{[i,L]} \mathbf{A}_{[i,L]}^T \right) \\
&= \sigma^2 \operatorname{trace} \left( \mathbf{A}_{[i,L-1]} \bar{\mathbf{W}}_L^\dagger (\bar{\mathbf{W}}_L^\dagger)^T \mathbf{A}_{[i,L-1]}^T \right) \\
&= \sigma^2 \operatorname{trace} \left( \mathbf{A}_{[i,L-1]} (\bar{\mathbf{W}}_L^T \bar{\mathbf{W}}_L)^{-1} \mathbf{A}_{[i,L-1]}^T \right) \\
&\leq \frac{\sigma^2}{\lambda_{\min}(\bar{\mathbf{W}}_L^T \bar{\mathbf{W}}_L)} \operatorname{trace} \left( \mathbf{A}_{[i,L-1]} \mathbf{A}_{[i,L-1]}^T \right) \\
&\leq \quad \cdots \\
&\leq \frac{\sigma^2}{\prod_{j=i+1}^L \lambda_{\min}(\bar{\mathbf{W}}_j^T \bar{\mathbf{W}}_j)} \operatorname{trace} \left( \mathbf{A}_{[i,i]} \mathbf{A}_{[i,i]}^T \right) \\
&= \frac{\sigma^2}{\prod_{j=i+1}^L \lambda_{\min}(\bar{\mathbf{W}}_j^T \bar{\mathbf{W}}_j)} \operatorname{trace} \left( (\bar{\mathbf{W}}_i^T \bar{\mathbf{W}}_i)^{-1} \right) \\
&\leq \frac{\sigma^2}{\prod_{j=i}^L \lambda_{\min}(\bar{\mathbf{W}}_j^T \bar{\mathbf{W}}_j)} s_i,
\end{aligned}
\tag{16}
$$

and this concludes the proof.

$\square$

## C    THEOREM 3: PROOF

*Proof.* We first recall the stability guarantee from Tropp (2006) for the basis-pursuit.

**Lemma 2** (Basis Pursuit Stability Tropp (2006)). *Let $\mathbf{x}^*$ be an unknown sparse representation with known cardinality of $\|\mathbf{x}^*\|_0 = s$, and let $\mathbf{y} = \mathbf{W}\mathbf{x}^* + \mathbf{e}$, where $\mathbf{W}$ is a matrix with unit-norm columns and $\|\mathbf{e}\|_2 \leq \epsilon$. Assume the mutual coherence of the dictionary $\mathbf{W}$ satisfies $s < 1/(3\mu(\mathbf{W}))$. Let $\hat{\mathbf{x}} = \arg\min_{\mathbf{x}} \frac{1}{2} \|\mathbf{y} - \mathbf{W}\mathbf{x}\|_2^2 + \lambda \|\mathbf{x}\|_1$, with $\lambda = 2\epsilon$. Then, $\hat{\mathbf{x}}$ is unique, the support of $\hat{\mathbf{x}}$ is a subset of the support of $\mathbf{x}^*$, and*

$$
\|\mathbf{x}^* - \hat{\mathbf{x}}\|_\infty < (3 + \sqrt{1.5})\epsilon.
\tag{17}
$$

In order to use the above lemma in our analysis we need to modify it such that $\mathbf{W}$ does not need to be column normalized and that the error is $\ell_2$- and not $\ell_\infty$-bounded. For the first modification we decompose a general unnormalized matrix $\mathbf{W}$ as $\tilde{\mathbf{W}}\mathbf{D}$, where $\tilde{\mathbf{W}}$ is the normalized matrix, $\tilde{\mathbf{w}}_i = \mathbf{w}_i / \|\mathbf{w}_i\|_2$, and $\mathbf{D}$ is a diagonal matrix with $d_i = \|\mathbf{w}_i\|_2$. Using the above lemma we get that

$$
\|\mathbf{D}(\mathbf{x}^* - \hat{\mathbf{x}})\|_\infty < (3 + \sqrt{1.5})\epsilon.
\tag{18}
$$

Thus, the error in $\hat{\mathbf{x}}$ is bounded by

$$
\|\mathbf{x} - \hat{\mathbf{x}}\|_\infty < \frac{(3 + \sqrt{1.5})}{\min_i \|\mathbf{w}_i\|_2} \epsilon.
\tag{19}
$$

Since Lemma 2 guarantees that the support of $\hat{\mathbf{x}}$ is a subset of the support of $\mathbf{x}^*$, we can conclude that

$$
\|\mathbf{x} - \hat{\mathbf{x}}\|_2 < \frac{(3 + \sqrt{1.5})}{\min_i \|\mathbf{w}_i\|_2} \epsilon \sqrt{s}.
\tag{20}
$$

Under the conditions of Theorem 3, we can use the above conclusion to guarantee that estimating $\mathbf{x}_L$ from the noisy input $\mathbf{y}$ using Basis-Pursuit must lead to a unique $\hat{\mathbf{x}}_L$ such that its support is a subset of that of $\mathbf{x}_L$. Also,

$$
\|\mathbf{x}_L - \hat{\mathbf{x}}_L\|_2 < \epsilon_L = \frac{(3 + \sqrt{1.5})}{\min_j \|\mathbf{w}_{L,j}\|_2} \epsilon_{L+1} \sqrt{s_L},
\tag{21}
$$

---

**Algorithm 5** Latent Pursuit: Last Layer Inversion

---

**Input:** $\mathbf{y} \in \mathbb{R}^n$, $K \in \mathbb{N}$, $\lambda_L \geq 0$, $\mu \in (0, \frac{2}{\ell})$, $\phi(\cdot)$ is $\ell$-smooth and strictly monotonic increasing.
**Initialization:** $\mathbf{u}^{(0)} \leftarrow \mathbf{0}$, $\mathbf{x}_L^{(0)} \leftarrow \mathbf{0}$, $t^{(0)} \leftarrow 1$.
**General step:** for any $k = 0, 1, \ldots, K$ execute the following:

1. $\mathbf{g} \leftarrow \mathbf{W}_L^T \phi' \left( \mathbf{W}_L \mathbf{x}_L^{(k)} \right) \left[ \phi \left( \mathbf{W}_L \mathbf{x}_L^{(k)} \right) - \mathbf{y} \right]$

2. $\mathbf{u}^{(k+1)} \leftarrow \mathrm{ReLU} \left( \mathbf{x}_L^{(k)} - \mu \cdot (\mathbf{g} + \lambda_L \mathbf{1}) \right)$

3. $t^{(k+1)} \leftarrow \frac{1 + \sqrt{1 + 4t^{(k)2}}}{2}$

4. $\mathbf{x}_L^{(k+1)} \leftarrow \mathbf{u}^{(k+1)} + \frac{t^{(k)} - 1}{t^{(k+1)}} (\mathbf{u}^{(k+1)} - \mathbf{u}^{(k)})$

**Return:** $\mathbf{x}_L^{(K)}$

---

where $\mathbf{w}_{L,j}$ is the $j$th column in $\mathbf{W}_L$, and $\epsilon_{L+1} = \epsilon \ell$ as $\phi^{-1}(\mathbf{y})$ can increase the noise by a factor of $\ell$.

Moving on to the estimation of the previous layer, we have that $\hat{\mathbf{x}}_L^{\hat{S}_L} = \mathbf{W}_{L-1}^{\hat{S}_L} \mathbf{x}_{L-1} + \mathbf{e}_L$, where $\|\mathbf{e}_L\|_2 \leq \epsilon_L$. According to Theorem 3 assumptions, the mutual coherence condition holds, and therefore, we get that the support of $\hat{\mathbf{x}}_{L-1}$ is a subset of the support of $\mathbf{x}_{L-1}$, $\hat{\mathbf{x}}_{L-1}$ is unique, and that

$$\|\mathbf{x}_{L-1} - \hat{\mathbf{x}}_{L-1}\|_2 < \epsilon_{L-1} = \frac{(3 + \sqrt{1.5})}{\min_j \left\| \mathbf{w}_{L-1,j}^{\hat{S}_L} \right\|_2} \epsilon_L \sqrt{s_{L-1}}. \tag{22}$$

Using the same technique proof for all the hidden layers results in

$$\|\mathbf{x}_i - \hat{\mathbf{x}}_i\|_2 < \epsilon_i = \frac{(3 + \sqrt{1.5})}{\min_j \left\| \mathbf{w}_{i,j}^{\hat{S}_{i+1}} \right\|_2} \epsilon_{i+1} \sqrt{s_i}, \text{ for all } i \in \{1, \ldots, L-1\}, \tag{23}$$

where $\mathbf{w}_{i,j}^{\hat{S}_{i+1}}$ is the $j$th column in $\mathbf{W}_i^{\hat{S}_{i+1}}$.

Finally, we have that $\hat{\mathbf{x}}_1^{\hat{S}_1} = \mathbf{W}_0^{\hat{S}_1} \mathbf{z} + \mathbf{e}_1$, where $\|\mathbf{e}_1\|_2 \leq \epsilon_1$. Therefore, if $\varphi = \lambda_{\min}((\mathbf{W}_0^{\hat{S}_1})^T \mathbf{W}_0^{\hat{S}_1}) > 0$, and

$$\hat{\mathbf{z}} = \arg\min_{\mathbf{z}} \frac{1}{2} \left\| \hat{\mathbf{x}}_1^{\hat{S}_1} - \mathbf{W}_0^{\hat{S}_1} \mathbf{z} \right\|_2^2. \tag{24}$$

Then,

$$\|\hat{\mathbf{z}} - \mathbf{z}\|_2^2 = \mathbf{e}_1^T \left( (\mathbf{W}_0^{\hat{S}_1})^T \mathbf{W}_0^{\hat{S}_1} \right)^{-1} \mathbf{e}_1 \leq \frac{1}{\varphi} \epsilon_1^2, \tag{25}$$

which concludes Theorem 3 guarantees.

$\square$

## D   DETAILS ON THE LATENT-PURSUIT ALGORITHM

Here we provide additional details on the Latent-Pursuit algorithm described in Section 5.

In order to estimate the last layer we aim to solve

$$\mathbf{x}_L = \arg\min_{\mathbf{x}} \frac{1}{2} \|\mathbf{y} - \phi(\mathbf{W}_L \mathbf{x})\|_2^2 + \lambda_L \mathbf{1}^T \mathbf{x}, \text{ s.t. } \mathbf{x} \geq \mathbf{0}. \tag{26}$$

For this goal we make use of FISTA (Beck & Teboulle, 2009) algorithm as described in Algorithm 5.

As describe in Section 5, for estimating the middle layers we aim to solve:

$$\mathbf{x}_i = \arg\min_{\mathbf{x}} \frac{1}{2} \left\| \mathbf{x}_{i+1}^{\hat{S}} - \mathbf{W}_i^{\hat{S}} \mathbf{x} \right\|_2^2 + \lambda_i \mathbf{1}^T \mathbf{x}, \text{ s.t. } \mathbf{x} \geq \mathbf{0}, \ \mathbf{W}_i^{\hat{S}^c} \mathbf{x} \leq \mathbf{0}. \tag{27}$$

Using the auxiliary variable $\mathbf{a} = \mathbf{W}_i^{\mathcal{S}^c}\mathbf{x}$ and the positive semidefinite matrix $\mathbf{Q} = \alpha\mathbf{I} - \mathbf{W}_i^{\mathcal{S}^T}\mathbf{W}_i^{\mathcal{S}} + \beta\mathbf{I} - \rho_i\mathbf{W}_i^{\mathcal{S}^c T}\mathbf{W}_i^{\mathcal{S}^c}$, we get that the Linearized-ADMM aims to solve:

$$\min_{\mathbf{x},\mathbf{a},\mathbf{u}} \frac{1}{2}\left\|\mathbf{x}_{i+1}^{\mathcal{S}} - \mathbf{W}_i^{\mathcal{S}}\mathbf{x}\right\|_2^2 + \lambda_i\mathbf{1}^T\mathbf{x} + \frac{\rho_i}{2}\left\|\mathbf{a} - \mathbf{W}_i^{\mathcal{S}^c}\mathbf{x} + \mathbf{u}\right\|_2^2 + \frac{1}{2}\left\|\mathbf{x} - \mathbf{x}^{(k)}\right\|_{\mathbf{Q}}, \text{ s.t. } \mathbf{x} \geq \mathbf{0},\ \mathbf{a} \leq \mathbf{0}. \tag{28}$$

This leads to an algorithm that alternates through the following steps:

$$\mathbf{x}^{(k+1)} \leftarrow \arg\min_{\mathbf{x}} \frac{\alpha}{2}\left\|\mathbf{x} - \left(\mathbf{x}^{(k)} - \frac{1}{\alpha}\mathbf{W}_i^{\mathcal{S}^T}\left(\mathbf{W}_i^{\mathcal{S}}\mathbf{x}^{(k)} - \mathbf{x}_{i+1}^{\mathcal{S}}\right)\right)\right\|_2^2 + \lambda_i\mathbf{x} + \tag{29}$$

$$\frac{\beta}{2}\left\|\mathbf{x} - \left(\mathbf{x}^{(k)} - \frac{\rho_i}{\beta}\mathbf{W}_i^{\mathcal{S}^c T}\left(\mathbf{W}_i^{\mathcal{S}^c}\mathbf{x}^{(k)} - \mathbf{a}^{(k)} - \mathbf{u}^{(k)}\right)\right)\right\|_2^2, \text{ s.t. } \mathbf{x} \geq \mathbf{0}.$$

$$\mathbf{a}^{(k+1)} \leftarrow \arg\min_{\mathbf{a}} \frac{\rho_i}{2}\left\|\mathbf{a} - \mathbf{W}_i^{\mathcal{S}^c}\mathbf{x}^{(k+1)} + \mathbf{u}^{(k)}\right\|_2^2, \text{ s.t. } \mathbf{a} \leq \mathbf{0}. \tag{30}$$

$$\mathbf{u}^{(k+1)} \leftarrow \mathbf{u}^{(k)} + \left(\mathbf{a}^{(k+1)} - \mathbf{W}_i^{\mathcal{S}^c}\mathbf{x}^{(k+1)}\right). \tag{31}$$

Thus, the Linearized-ADMM algorithm, described in Algorithm 2 is guaranteed to converge to the optimal solution of Equation 7.

After recovering all the hidden layers, we aim to estimate the latent vector $\mathbf{z}$. For this inversion step we adopt a MAP estimator as described in Section 5:

$$\mathbf{z} = \arg\min_{\mathbf{z}} \frac{1}{2}\left\|\mathbf{x}_1^{\mathcal{S}} - \mathbf{W}_0^{\mathcal{S}}\mathbf{z}\right\|_2^2 + \frac{\gamma}{2}\left\|\mathbf{z}\right\|_2^2, \text{ s.t. } \mathbf{W}_0^{\mathcal{S}^c}\mathbf{z} \leq \mathbf{0}, \tag{32}$$

with $\gamma > 0$. In fact, this problem can be solved by a similar Linearized-ADMM algorithm described above, expect for the update of $\mathbf{x}$ (Equation 29), which becomes:

$$\mathbf{z}^{(k+1)} \leftarrow \arg\min_{\mathbf{z}} \frac{\alpha}{2}\left\|\mathbf{z} - \left(\mathbf{z}^{(k)} - \frac{1}{\alpha}\mathbf{W}_0^{\mathcal{S}^T}\left(\mathbf{W}_0^{\mathcal{S}}\mathbf{z}^{(k)} - \mathbf{x}_1^{\mathcal{S}}\right)\right)\right\|_2^2 +$$
$$\frac{\beta}{2}\left\|\mathbf{z} - \left(\mathbf{z}^{(k)} - \frac{\rho_i}{\beta}\mathbf{W}_0^{\mathcal{S}^c T}\left(\mathbf{W}_0^{\mathcal{S}^c}\mathbf{z}^{(k)} - \mathbf{a}^{(k)} - \mathbf{u}^{(k)}\right)\right)\right\|_2^2 + \frac{\gamma}{2}\left\|\mathbf{z}\right\|_2^2. \tag{33}$$

Equivalently, for the latent vector $\mathbf{z}$, the first step of Algorithm 2 is changed to to:

$$\mathbf{z}^{(k+1)} \leftarrow \frac{1}{\alpha + \beta + \gamma}\left((\alpha+\beta)\mathbf{z}^{(k)} - \mathbf{W}_0^{\mathcal{S}^T}(\mathbf{W}_0^{\mathcal{S}}\mathbf{z}^{(k)} - \mathbf{x}_1^{\mathcal{S}}) - \rho_1\mathbf{W}_0^{\mathcal{S}^c T}(\mathbf{W}_0^{\mathcal{S}^c}\mathbf{z}^{(k)} - \mathbf{a}^{(k)} - \mathbf{u}^{(k)})\right).$$

## E  INVERSION RESULTS FOR TRAINED NETWORKS

Here we provide detailed results for the various inversion experiments described in Section 6.2.

### E.1  CLEAN IMAGES

Figure 4 demonstrates the reconstruction error for all the layers when inverting clean images. In Figures 5 and 6 we demonstrate successful and failure cases of the gradient-descent algorithm and compare them to our approach.

### E.2  RANDOM MASK INPAINTING

Figures 7-9 demonstrate the performance of our approach compared to gradient descent in terms of SNR and image quality respectively for the randomly-generated mask experiment.

### E.3  NON-RANDOM MASK INPAINTING

Figures 10-12 demonstrate the performance of our approach compared to gradient descent in terms of SNR and image quality respectively for the non-random mask experiment.

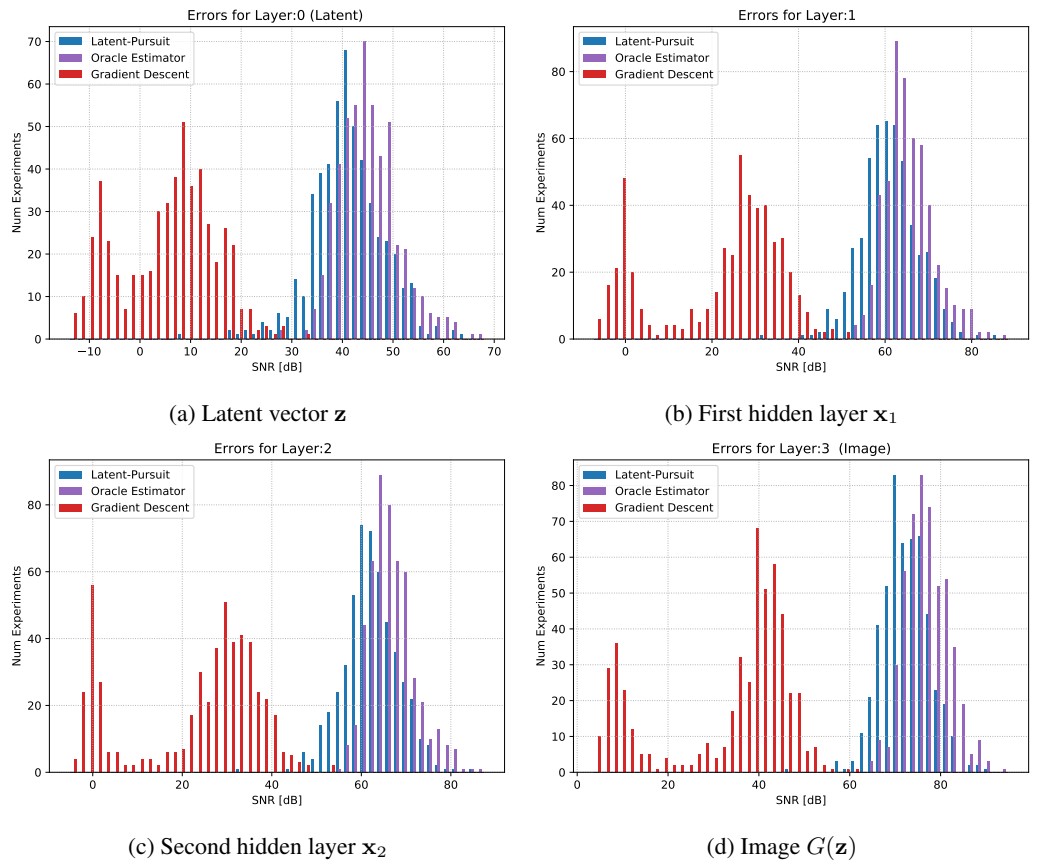

Figure 4: Trained 3-Layers Model: Reconstruction error for all the layers for 512 clean images. As can be observed, the Latent-Pursuit almost mimics the oracle, and outperforms gradient descent.

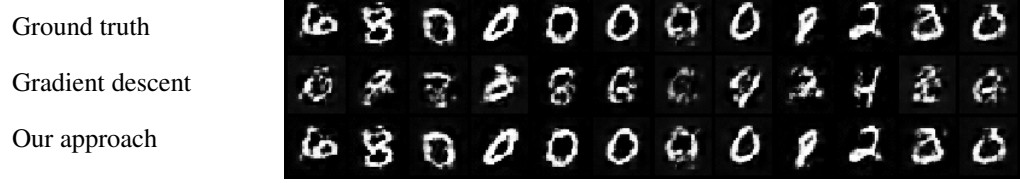

Figure 5: Reconstruction failures of gradient descent on clean images.

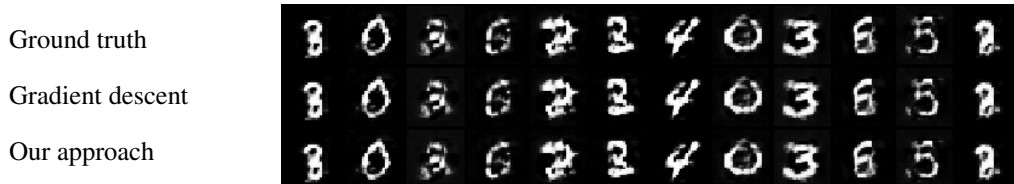

Figure 6: Successful reconstructions of gradient descent on clean images.

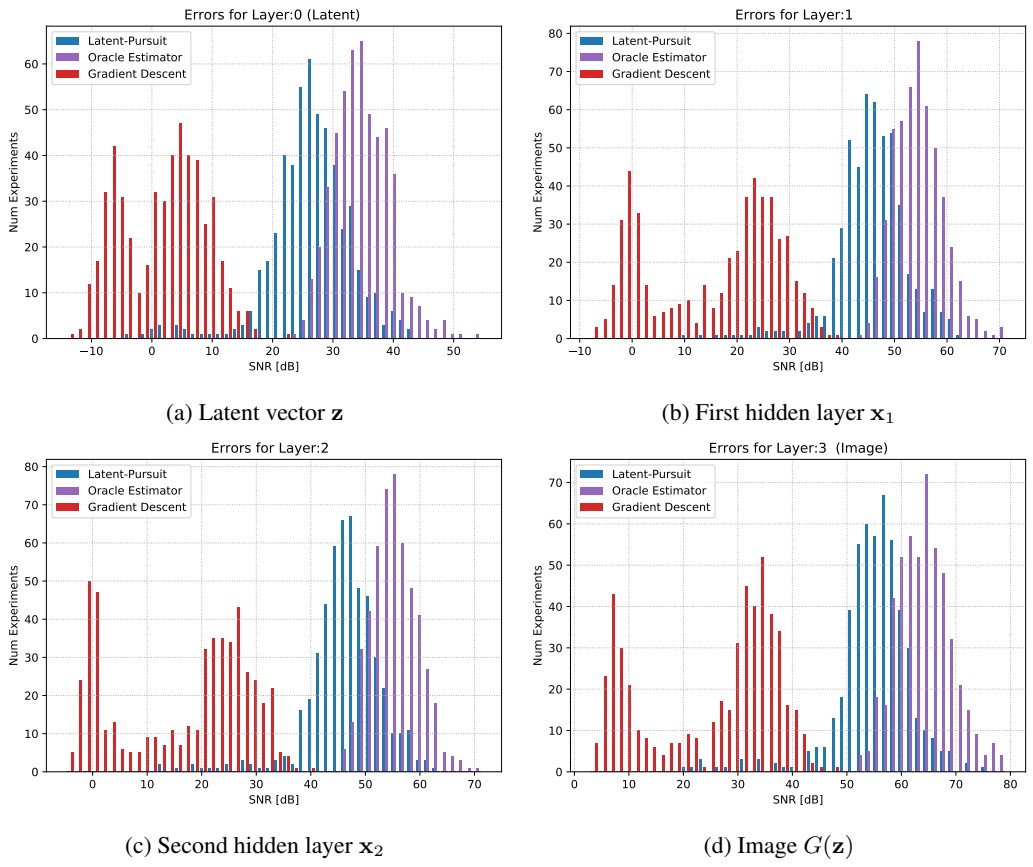

Figure 7: Random mask inpainting reconstruction error for all the layers on a trained generator.

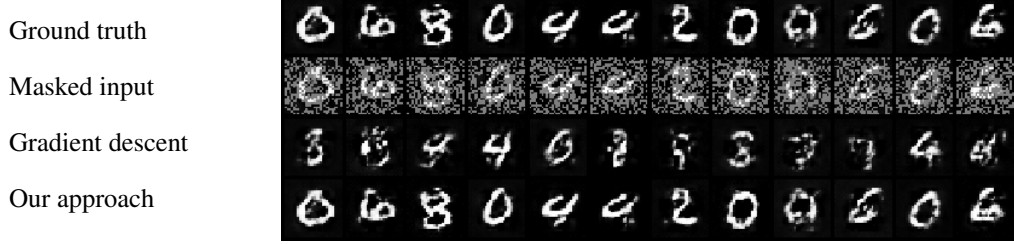

Figure 8: Random mask inpainting gradient descent failed reconstructions.

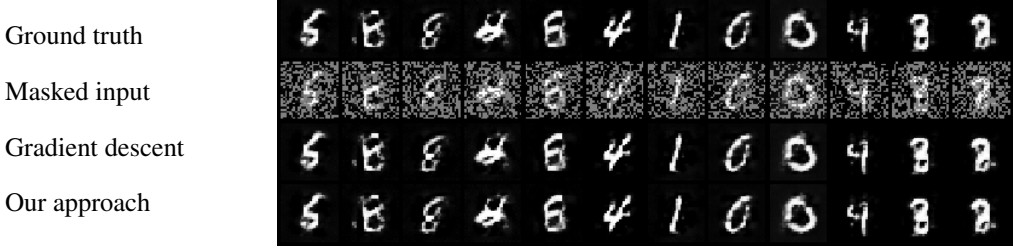

Figure 9: Random mask inpainting gradient descent successful reconstructions.

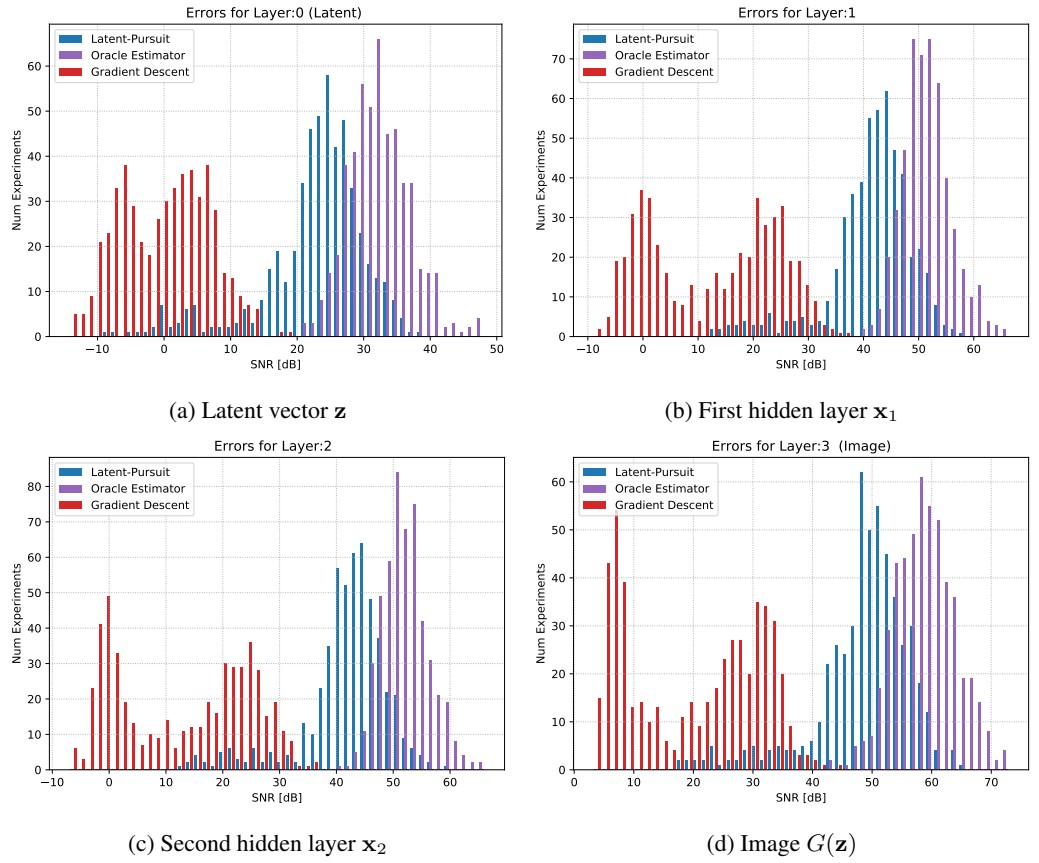

(a) Latent vector $\mathbf{z}$

(b) First hidden layer $\mathbf{x}_1$

(c) Second hidden layer $\mathbf{x}_2$

(d) Image $G(\mathbf{z})$

Figure 10: Half image mask inpainting reconstruction error for all the layers.

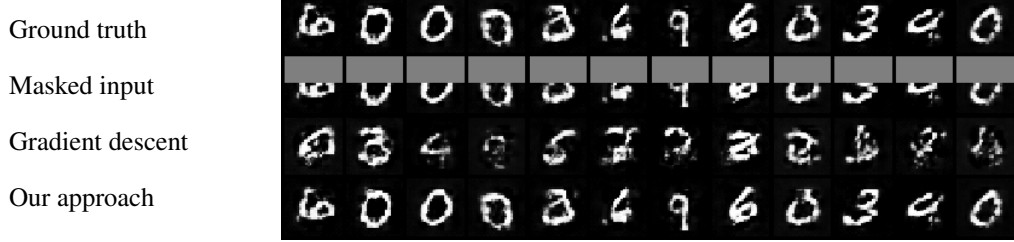

Figure 11: Half image mask inpainting gradient descent failed reconstructions.

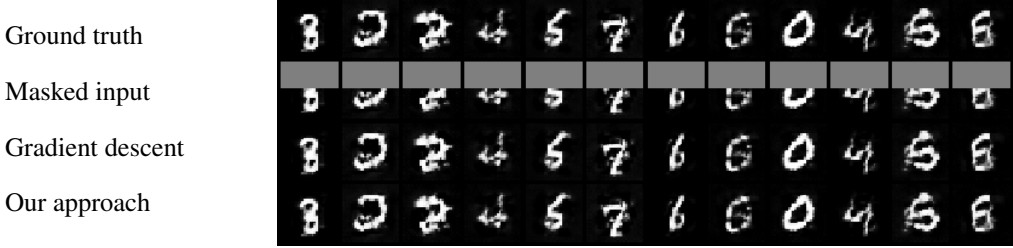

Figure 12: Half image mask inpainting gradient descent successful reconstructions.

