# OpenReview forum: "On the Inversion of Deep Generative Models"
_ICLR.cc/2021/Conference — Reject_

### Official Review · AnonReviewer3 · 2020-10-25
**Theoretical Recovery Results for ReLU network inversion without random weight assumptions**

**Rating:** 7
**Confidence:** 4

**Review:**

In this submission, the authors study the inversion of ReLU networks (where the output of the network is subject to an invertible activation function).  This is an important task, for example for inverse problems using generative priors.  The authors introduce spark-based conditions for the invertibility of each layer of the network, leveraging sparsity that is induced by ReLUs.  The authors also introduce a novel layer wise inversion algorithm and provide provable recovery guarantees in both noisy and noiseless settings.  Empirical results demonstrate the superiority of the proposed algorithm relative to baselines for inversion in particular parameter regimes.

Theorem 1 on invertibility/uniqueness is a significant contribution in that it permits an invertibility guarantee for neural networks without random weight assumptions.  Previous works involved weights coming from particular distributions.  Theorem 3 is significant because it establishes exact-noiseless recovery or stable-noisy recovery of an inversion algorithm without assuming random wights or growth rates on the network widths.  The authors present a clear and sophisticated algorithmic implementation, where different numerical approaches are used at each layer of an inversion process.

There are a few minor issues the authors should fix upon revision:
(1) In Section 2, the authors should state that n_{L+1}=n.
(2) It is not clear whether the authors are simply restating Theorem 2 from Foucart and Rauhut or whether this is a novel contribution.  The connection between the Theorem and the preceding paragraph needs to be clarified.  It appears to the reviewer that Theorem 1's conditions in the random case are more restrictive than Theorem 2's conditions (by a factor of 2).
(3) In Section 1.1 paragraph 2, the authors write that they relax expansion assumptions as to rely only on the expansion of the number of non-zero elements.  The commentary on this point during the results sections could be more thorough.  The lines above Theorem 2 indicate that in the random case, the sparsity levels must grow by approximately a factor of two at each layer.  The paper does not say so, but this condition requires that the network widths are larger than a particular geometric sequence with growth rate 2.  I believe this statement also holds true in the non-random case as well.  Please add commentary to this effect after Theorem 1.
(4) Defn 3: coherent -> coherence
(5) Theorem 3: epsilon_i isn't defined. It is defined within Algorithm 1, but this should be clearly pointed out.

---

> ### Author Response · Authors · 2020-11-13
> **Authors response to reviewer 3**
>
> We thank the reviewer for the detailed review and for the constructive comments made, which helped in improving our paper. We provide our answers below:
>
> > “In Section 2, the authors should state that n_{L+1}=n.”
>
> Thank you for noticing this issue. We now state it explicitly at the beginning of Section 2.
>
> > "It is not clear whether the authors are simply restating Theorem 2 from Foucart and Rauhut or whether this is a novel contribution. The connection between the Theorem and the preceding paragraph needs to be clarified…”
>
> Thank you for this comment. Indeed, a direct corollary of Theorem 1 for random weights would be more restrictive (by a factor of 2) compared to Theorem 2. Nevertheless, to relax these conditions for the random setting, we utilize Foucart & Rauhut’s theorem. This theorem considers typical sparse representation with random dictionaries and states that a sparse representation is unique as long as its cardinality is smaller than the signal dimension (with probability 1). The proof for Theorem 2 is similar to Theorem 1’s proof while replacing Donoho & Elad’s Spark-based theorem with Foucart & Rauhut’s. Following this comment, we clarified the connection between these theorems before stating Theorem 2.
>
> > “In Section 1.1 paragraph 2, the authors write that they relax expansion assumptions as to rely only on the expansion of the number of non-zero elements. The commentary on this point during the results sections could be more thorough. The lines above Theorem 2 indicate that in the random case, the sparsity levels must grow by approximately a factor of two at each layer. The paper does not say so, but this condition requires that the network widths are larger than a particular geometric sequence with growth rate 2. I believe this statement also holds true in the non-random case as well. Please add commentary to this effect after Theorem 1.”
>
> For the random setting, the lines before theorem 2 (which are a direct corollary of theorem 1) indeed suggest that an expansion growth of factor 2 is required. However, using Foucart & Rauhut’s theorem, we show in theorem 2 that any (constant) expansion in the number of non-zero elements is sufficient to assure uniqueness.  The experiments in Section 6.1 demonstrate our theory. Here, the network’s output $x$ is dense, and its inversion is successful even when the size of the middle layer is larger than the image size, i.e. $n_1 > n$. This occurs since the number of non-zero elements increases $\|x_1\|_0 < \|x\|_0$. The inversion holds even when the cardinality increases by a factor smaller than two, e.g. $n_1=800$, or equivalently $\|x_1\|_0=400$, while $n=\|x\|_0=625$.
> Regarding Theorem 1 and the non-random setting, we agree with the reviewer’s comment.
> We added a discussion on this matter in Section 3 before and after theorem 2. Thank you for helping us improve our exposition.
>
> > “Defn 3: coherent -> coherence
> > Theorem 3: epsilon_i isn't defined…”
>
> Fixed. Thank you.

---

### Official Review · AnonReviewer4 · 2020-10-29
**A theorem providing sufficient conditions of NN invertibility and a pursuit inversion method on top.**

**Rating:** 3
**Confidence:** 5

**Review:**

Clarity:
The paper is relatively clearly written except for the experimental section.

Originality:
The content of this paper is not novel.

Significance:
The claimed contribution of this paper is not significant.

Cons:

This work tries to rigorously characterize the conditions for a rather intuitive task (reconstruction). There are a few serious issues around it.

First, without having the learned weights satisfying certain conditions exactly, the theorems (e.g. Thm1) are way too conservative, i.e. the bounds are too loose in order to catch all corner cases. In addition, in reality, the activations are not necessarily sparse, especially for generative models. The combination of these caveats make these theorems' indications of little meaning.

Second, in reality, we use much more complex architecture (CNN, attention, ReLU, Normalization) but these theoretical works can't easily go beyond simple MLP with ReLU, making these results even less meaningful.

Thirdly, despite all the above, it is hard for me to reason why the theorems are significant in anyway in the context of ML, e.g. why uniqueness is important. I can understand the significance of existence and uniqueness in the context of differential equation but not ML.

Other comments:
What is "sparse representation theory"? I know "sparse representation", and "representation theory", but not sure about "sparse representation theory".

Lack of novelty: the proofs are direct extensions of existing compress sensing literature.

Experiments:
-the experimental setups are quite contrived and not optimized for model performance but for the papers' plots. E.g. a larger scale b etter designed MLP can model MNIST dataset quite well (e.g. 4 layers of 1024x1024 with LeakyReLU)
-the evaluation metric is suboptimal. Eventually, we want to measure the recovered signal, not the latent code error.

---

> ### Author Response · Authors · 2020-11-13
> **Authors response to reviewer 4 - 3/3**
>
> (3/3)
>
> > “Lack of novelty: the proofs are direct extensions of existing compress sensing literature.”
>
> We respectfully disagree with the reviewer regarding the lack of novelty in our work. This work is the first to establish the connection between sparse representation and deep generative models. This is a novel approach for solving the discussed inverse problem and leads to theorems that relax previously known conditions on the invertibility of random deep generative models and the first one to provide conditions on the invertibility of non-random models.
>
> > “the experimental setups are quite contrived and not optimized for model performance but for the papers' plots… Eventually, we want to measure the recovered signal, not the latent code error.“
>
> As depicted at the beginning of Section 6, the experimental section aims to fulfill two goals: (i) demonstrate our theoretical claims in the random setting, e.g. Theorem 2, and (ii) empirically demonstrate the effectiveness of our algorithm for solving the inverse problem in Eq. (1) compared to gradient descent. Indeed, we measure the success of the inversion process by evaluating it in the image domain, i.e. the rightmost column in Figure 2 and Appendixes E1-E3, which empirically show the advantage of our approach over gradient descent. In addition, we enrich our analysis by also showing the intermediate results on the recovered latent vectors, exposing the inner-works and tendencies of our algorithms and the competing methods.
>
> [1] - Hand, Paul et al. Phase retrieval under a generative prior. NeurIPS, 2018.
>
> [2] - Hand, Paul et al. Global guarantees for enforcing deep generative priors by empirical risk. IEEE Transactions on Information Theory, 2019.
>
> [3] - Latorre, Fabian et al. Fast and provable admm for learning with generative priors. NeurIPS, 2019.
>
> [4] - Lei, Qi et al. Inverting deep generative models, one layer at a time. NeurIPS, 2019.
>
> [5] - Radford, Alec et al. "Unsupervised representation learning with deep convolutional generative adversarial networks." (2015).
>
> [6] - Karras, Tero, et al. "Progressive growing of gans for improved quality, stability, and variation." (2017).

---

> ### Author Response · Authors · 2020-11-13
> **Authors response to reviewer 4 - 2/3**
>
> (2/3)
>
> We now address each of the reviewer's comments:
>
> > “This work tries to rigorously characterize the conditions for a rather intuitive task (reconstruction).“
>
> The reviewer states that the task we study, i.e. inverting a ReLU activated generative model, is rather intuitive. In fact, as shown by Lei et al. 2019 Theorems 1-2, this task is non-convex and NP-complete in several settings. While previous works usually solve it using gradient-based methods, it is unknown when these methods succeed or fail. Note that several papers [1-4] studied this particular task, providing theoretical claims only when the weights are assumed to be random.
> Moreover, as depicted in Figures 2 and 3, there are various cases in which gradient descent fails to reconstruct the image and to recover the latent vector.
> Our work provides theoretical conditions for when this problem is practically solvable, as well as suggesting algorithms with recovery guarantees for this task. We added a description highlighting the hardness of this problem in the introduction after Eq. (1).
>
> > “...without having the learned weights satisfying certain conditions exactly, the theorems are way too conservative…”
>
> Indeed, our inversion theorems rely on the properties of the generator weights. While the reviewer claims that these are too conservative, our results are the first ones to provide invertibility guarantees for general non-random (trained) networks, as written in the main contributions paragraph in Section 1 and after Theorem 1. Furthermore, Theorem 2, which addresses the previously-studied random setting, provides relaxed conditions compared to previous work.
>
>
> > “In addition, in reality, the activations are not necessarily sparse, especially for generative models.”
>
> Regarding the concern on the sparsity level of the hidden layers, please note that the theorems rely on the growth of the cardinality over the layers. The common practice in deep generative models is to increase the size of the hidden layers across the model’s depth, making the conditions of the theorems reasonable. For example, if the layers expand by a factor of 2 (as in DCGAN [5] and PGAN [6]), then the hidden layers need only to maintain a similar percentage of nonzero elements to assure uniqueness, e.g. 50%. Hence, the representations themselves should not be extremely sparse. Beyond this, please recall that the representations are obtained after a ReLU step, forcing zero values, and thus pushing towards sparsity.
>
> > “Second, in reality, we use much more complex architecture (CNN, attention, ReLU, Normalization)...”
>
> Note that our models do contain batch-normalization and ReLU activations and support other normalization techniques. We agree that the provided theory is less applicable to attention-based models. That said, building on this work, an extension to the convolutional setting is plausible and underway.
>
> > “... these theoretical works can't easily go beyond simple MLP with ReLU, making these results even less meaningful.”
>
> Indeed, at the current state of things, there exists a large gap between theoretical understanding and practical achievements in the field of deep learning. Our work and previous literature focus on the effort of closing this gap. Here, we target the actively studied problem of inverting deep generative models and significantly relax previously known conditions for their invertibility.
> In addition, as we demonstrate throughout Section 6, our proposed algorithms are in fact superior to gradient descent (the current standard go-to solution) in several settings.
>
> > “Thirdly, ... why the theorems are significant in anyway in the context of ML, e.g. why uniqueness is important. I can understand
> > the significance of existence and uniqueness in the context of differential equation but not ML.”
>
> Solving Eq. (1) is crucial for various applications, e.g. compressed sensing, image inpainting, and other inverse problems. This work and previous literature seek to answer two questions: when a solution to Eq. (1) is obtainable, and how to retrieve such a desired solution. Our work provides a constructive answer to both questions: Using the provided uniqueness theorems, we are able to guarantee the recovery of the solution for this highly non-convex problem. In addition, we offer guaranteed recovery algorithms.

---

> ### Author Response · Authors · 2020-11-13
> **Authors response to reviewer 4 - 1/3**
>
> (1/3)
>
> We thank the reviewer for assessing our work and provide our answers to his/her comments below.
>
> Before addressing each of the reviewer’s comments, we would like to note that most of the above comments contend any paper in this line of work (e.g. [1-4]), and not only this paper. This includes the following claims (by order):
>
> * “a rather intuitive task.”
> * “bounds are too loose in order to catch all corner cases.”
> * “in reality, we use much more complex architecture.”
> * “it is hard for me to reason why the theorems are significant in anyway in the context of ML.”
> * “the experimental setups are quite contrived and not optimized for model performance.”
>
> Therefore, we would like to start by clarifying the goal of this line of work.
>
> Inverting generative models is crucial in various applications, e.g. compressed sensing, image inpainting, image denoising, and other inverse problems. In recent years, traditional theoretically-sound models have been replaced with neural networks, bringing empirical success. Unfortunately, there are various cases in which the recovery of the latent vector and the reconstruction of the image fail. The reviewer may find several examples for these failures in Figures 2, 3, and Appendices E1-E3. The goals of this line of work, which this paper belongs to, are two-fold: First, defining when such an inversion can or cannot be performed, and second, proposing algorithms that solve this task with provable guarantees. We, as well as the other reviewers, believe that our work contributes to both these directions significantly.

---

### Official Review · AnonReviewer1 · 2020-10-29

**Rating:** 6
**Confidence:** 3

**Review:**

Summary

This paper proposes the conditions for the invertibility of deep generative models, and two pursuit algorithms for inverting them. The idea is based on sparse representation theory, applied on layer-by-layer inversion. The authors claim this to be the first work that provides provable guarantees of inversion for general non-random neural networks.

This is a very solid piece of work, with significant contributions on both the theoretical and practical sides. However, the impact of this work might be limited by the sparsity assumption, which is central to this work. It is unclear whether this holds in more general settings, especially for very deep models trained on large scale datasets.

Pros
1. It is overall clearly written and well-structured.
2. This work is built on a solid theoretical foundation. It provides interesting connections between neural networks and signal processing / convex optimisation.
3. This work relaxes the assumptions in previous work [1], which makes it more relevant in practice.

Cons
1. It is unclear how well does the sparsity assumption in theorem 1 hold in practice for trained models, which may limit the impact of this work.
2. The claims of provable guarantees for inversion is exaggerated, as it only applies to Algorithm 1, which relies on an oracle to provide the true supports of all hidden layers. There is no such guarantee for the more general Algorithm 3.
3. The experiments are limited to toyish and simple dataset. They are unlikely to capture the statistics of e.g., natural images that are practically more relevant.

Other comments and questions:

1. How to reconcile the difference between the special case of theorem 1 with random matrices (the paragraph above theorem 2) and theorem 2?
2. At the end of page 2, the complementary S^c_{i+1} is used before it is defined after eq. 5.
3. What is the distribution of the random input signal used in section 6.1? How is it chosen?
4. It might be helpful to summarise the sparsity assumptions from theorem 1 in the abstract or introduction, so that the limitation of the analysis and method is more clear for readers.
5. For gradient descent baselines, it worth considering (at least discussing) algorithms that jointly train the model with gradient descent (e.g.,  [2], [3]).

[1] Inverting Deep Generative models, One layer at a time, Lei et al. 2019
[2] Optimizing the Latent Space of Generative Networks, Bojanowski et al. 2017
[3] Deep Compressed Sensing, Wu et al. 2019

---

> ### Author Response · Authors · 2020-11-13
> **Authors response to reviewer 1**
>
> We thank the reviewer for the detailed review and for the constructive comments made, which helped in improving our paper. We provide our answers below (by order):
>
> > “This is a very solid piece of work… However, the impact of this work might be limited by the sparsity assumption...”
> > “It is unclear how well does the sparsity assumption in theorem 1 hold in practice for trained models, which may limit the impact of this work.”
>
> Indeed, our theorems rely on the sparsity level of the layers. More accurately, however, they rely on the growth of the cardinality over the layers. This is an important distinction since the common practice in deep generative models is to increase the size of the hidden layers across the depth. For example, if the layers expand by a factor of 2 (as in DCGAN [1] and PGAN [2]), then the hidden layers need to maintain a similar percentage of nonzeros to assure uniqueness. Hence, the representations themselves should not be extremely sparse.
> Furthermore, in contrast with previous work, e.g. [3-5], our theorems allow for a decrease in the size of the layers themselves as long as the cardinality increases. This is especially needed in the last hidden layer, which is typically larger than the image size. This attribute is demonstrated in Figure 2 in which we obtain a successful inversion even when the size of the middle layer is larger than the output size, e.g. $n_1=800$ and $n=625$. Therefore, as the reviewer mentioned, the provided theorems are much less restrictive than previous work.
> We added a discussion along these lines in the paragraphs proceeding Theorem 2.
>
> > “The claims of provable guarantees for inversion is exaggerated, as it only applies to Algorithm 1, which relies on an oracle to provide the true supports of all hidden layers.”
>
> We emphasize that Algorithm 1 does not rely on knowing the oracle supports. However, the algorithm does include hyperparameters $\lambda_i$ and, as Theorem 3 states, these should be set correctly, relying on the cardinality of the ground-truth representations (and not the actual supports). That said, in practice, these hyperparameters could be tuned using a search procedure without a-priori knowledge regarding the cardinality of the representation layers.
>
> > “...There is no such guarantee for the more general Algorithm 3.”
>
> As for Algorithm 3, we agree that it comes with no theoretical guarantees (although we conjecture that it cannot perform worse than Algorithm 1, as indeed our experiments show). This has now been stated in the abstract. Indeed, providing theoretical guarantees for a pursuit algorithm with non-negativity constraints is an intriguing open question for future research.
>
> > “The experiments are limited to toyish and simple dataset... “
>
> The experiments reported in the paper follow the ones in [3]. The choice we have made to work with MNIST comes from the fact that all our analysis assumes fully-connected generative models, thus restricting our tests to low-dimensional signals. Our future work aims to expand these results to the convolutional sparse coding model, thereby enabling the treatment of convolutional layers, which is more relevant for natural image synthesis. We comment about this in Section 6.2.
>
> > “How to reconcile the difference between the special case of theorem 1 with random matrices … and theorem 2?”
>
> Thank you for this question. The description that comes before Theorem 2 is a direct corollary of Theorem 1 for the random setting. Then, in Theorem 2 itself, we further strengthen our claims by utilizing Foucart & Rauhut’s theorem. We revised the paragraphs preceding and following Theorem 2. We believe the difference is clearer now.
>
> > “At the end of page 2, the complementary S^c_{i+1} is used before it is defined after eq. 5.”
>
> Added to the notations paragraph. Thank you.
>
> > “What is the distribution of the random input signal used in section 6.1? How is it chosen?”
>
> In all our experiments the latent vector is Gaussian. We added this information in Section 6.
>
> > “It might be helpful to summarise the sparsity assumptions from theorem 1 in the abstract or introduction, so that the limitation of the analysis and method is more clear for readers.”
>
> We agree with this comment, and accordingly, we added a concise description of our bounds and conditions in the “main contributions” paragraph in the Introduction Section.
>
> > “For gradient descent baselines, it worth considering (at least discussing) algorithms that jointly train the model with gradient descent.”
>
> Thank you for referring our attention to this line of work. We added a discussion regarding these (and similar) papers in Section 1.1. Note that the main difference between these papers and our work is that we do not assume or require anything in the training procedure itself, and only rely on the weights of the final generative model.

---

> ### Author Response · Authors · 2020-11-13
> **Authors response to reviewer 1  - references**
>
> [1] - Radford, Alec et al. "Unsupervised representation learning with deep convolutional generative adversarial networks." (2015).
>
> [2] - Karras, Tero, et al. "Progressive growing of gans for improved quality, stability, and variation." (2017).
>
> [3] - Lei, Qi et al. Inverting deep generative models, one layer at a time. NeurIPS, 2019.
>
> [4] - Hand, Paul et al. Phase retrieval under a generative prior. NeurIPS, 2018.
>
> [5] - Hand, Paul et al. Global guarantees for enforcing deep generative priors by empirical risk. IEEE Transactions on Information Theory, 2019.

---

### Decision · Program_Chairs · 2021-01-07
**Final Decision**

**Decision:**

Reject

**Comment:**

The reviews are a bit mixed, so the AC independently examined the submission as well. While the authors' response helped clarifying some issues, the draft would still need some major revision, especially the motivation and the experiment part. Here are some concrete suggestions:

(a) The problem definition in Eq (1) is already problematic, as the authors term the least squares reconstruction as an "inversion." When exact reconstruction is not possible, one needs to question the choice of the 2-norm here: why 2-norm and what happens if we change the norm? How does this choice of norm enter the proof and the theorem statements? The uniqueness, as pointed out by one of the reviewers, requires further elaborations: It is not clear that for applications like image denoising or impainting it is necessary to have a unique latent vector, if all one cares about is good reconstruction.

(b) The authors need to explain the motivation of the "inversion" problem better, in the context of generative models. In generative models, G pushes a distribution p on R^{n_0} to q on R^n. If both p and q have nontrivial support and n_0 < n, the map G can be highly irregular and the reconstruction problem can be very ill-conditioned. Putting assumptions on G (such as incoherence) might heavily constrain what kind of distributions q we can learn. This trade-off was hardly discussed. In fact, if we take n_0 = n, then there exist normalizing flows that can learn any distribution q and that can be trivially inverted. If one is interested in inversion, why not use a universal normalizing flow?

(c) The authors mentioned a few possible applications (image denoising, compressed sensing, image inpainting) of the inversion problem, but none of them (to my best knowledge) relies on modelling the underlying distribution. There are also classic algorithms for each of these applications. It would be much more convincing if the authors could explain why a deep generative model is advantageous for these applications and compare the proposed algorithms on standard benchmarks of these applications. The AC agrees with the reviewers that the current experiments are a bit toy-ish (which is not wrong by itself but does require a bit more elaboration when the motivation is in question).

(d) The significance of Theorem 1 requires further elaboration. How does one verify its conditions? How often does a trained network satisfy these conditions? How do these conditions restrict the expressiveness of the underlying model? Some of these questions were asked in the reviews but regrettably the authors largely dismissed them. Results like Theorem 2 also require further clarification: what does random weight mean for a generative model? The authors seemed to only care about inversion while completely ignored the expressiveness of a generative model. As a trivial example, one could take a linear network, with regularization we can always invert (in the sense of the authors' definition) the signal. Why is this any different from (if not better than) the authors' results?